# Ribonucleotide reductase subunit switching in hepatoblastoma drug response and relapse

Anthony Brown [1], Qingfei Pan[2], Li Fan[1], Emilie Indersie [3], Cheng Tian[1], Nikolai Timchenko[4], Liyuan Li[1], Baranda S. Hansen [5], Haiyan Tan[6], Meifen Lu[7], Junmin Peng [8], Shondra M. Pruett-Miller [5], Jiyang Yu [2], Stefano Cairo [3] & Liqin Zhu [1]✉

Prognosis of children with high-risk hepatoblastoma (HB), the most common pediatric liver cancer, remains poor. In this study, we found ribonucleotide reductase (RNR) subunit M2 (*RRM2*) was one of the key genes supporting cell proliferation in high-risk HB. While standard chemotherapies could effectively suppress RRM2 in HB cells, they induced a significant upregulation of the other RNR M2 subunit, RRM2B. Computational analysis revealed distinct signaling networks RRM2 and RRM2B were involved in HB patient tumors, with RRM2 supporting cell proliferation and RRM2B participating heavily in stress response pathways. Indeed, RRM2B upregulation in chemotherapy-treated HB cells promoted cell survival and subsequent relapse, during which RRM2B was gradually replaced back by RRM2. Combining an RRM2 inhibitor with chemotherapy showed an effective delaying of HB tumor relapse in vivo. Overall, our study revealed the distinct roles of the two RNR M2 subunits and their dynamic switching during HB cell proliferation and stress response.

[1] Department of Pharmacy and Pharmaceutical Sciences, St. Jude Children's Research Hospital, Memphis, TN, USA. [2] Department of Computational Biology, St. Jude Children's Research Hospital, Memphis, TN, USA. [3] XenTech, Paris, France. [4] Department of Surgery, Cincinnati Children's Hospital Medical Center and Department of Surgery, University of Cincinnati, Cincinnati, OH, USA. [5] Department of Cell and Molecular Biology and Center for Advanced Genome Engineering, St. Jude Children's Research Hospital, Memphis, TN, USA. [6] Center for Proteomics and Metabolomics, St. Jude Children's Research Hospital, Memphis, TN, USA. [7] Center for Comparative Pathology Core, St. Jude Children's Research Hospital, Memphis, TN, USA. [8] Departments of Structural Biology and Developmental Neurobiology, St. Jude Children's Research Hospital, Memphis, TN, USA. ✉email: Liqin.zhu@stjude.org

Hepatoblastoma (HB) is a rare type of primary liver cancer that only affects very young children[1]. Although accounting for only 0.5-2% of all cancer cases in children[2], HB has the largest increase in incidence among childhood cancers in the United States and worldwide[3]. Most HB tumors are sensitive to chemotherapy and children with HB have an excellent overall five-year survival of >80%. But for children diagnosed with high-risk HB, this number drops to below 40% even with multidisciplinary therapies including surgery, chemotherapy, and radiotherapy[4]. Studies in many adult solid tumors have found that tumor cells can develop drug resistance during the course of treatment via various mechanisms as part of their adaption to stress conditions[5,6]. This raised an intriguing question that whether HB cells in high-risk tumors, with their known cellular and molecular heterogeneity and plasticity[7], can similarly evoke a self-defense machinery when treated by anti-cancer drugs to increase their chance of survival.

Recent work in our lab using HB mouse and organoid models[8], patient-derived xenografts (PDX) and primary patient samples revealed a significant upregulation of ribonucleotide reductase (RNR) in high-risk HB. RNR is the sole enzymatic complex in mammalian cells that converts ribonucleoside diphosphate (NDP) to deoxy-NDP (dNDP)[9]. It plays a critical role in regulating the total rate of DNA synthesis during cell division and DNA repair[10–12]. RNR is a heterodimeric tetramer composed of two large RNR subunit M1 (RRM1) and two small RNR subunit M2 (RRM2)[13]. RRM2 contains the catalytic domain of RNR and is tightly cell-cycle regulated[14]. Therefore, it is not surprising that RRM2 upregulation has been found in many adult cancers[15–19]. There is another low-expressing RNR M2 subunit, RNR subunit M2B (RRM2B), which has a lower catalytic activity than RRM2 and is not cell-cycle regulated[20]. It has been found that RRM2B can be induced in a p53-dependent manner under certain stress conditions and becomes the dominant M2 subunit to support DNA repair[21,22]. Since there is little known about the involvement of RNR in pediatric cancer, we decided to investigate RNR dynamics regarding to its role in HB progression and drug response.

## Results

### RRM2, not RRM2B, is associated with the poor prognosis of HB mouse and patient tumors.

We previously generated a HB mouse model by targeting a population of *Prom1*-expressing neonatal liver progenitors with an activating *Notch* mutation, NICD (Notch intercellular domain), *Prominin1*$^{CreERT2}$; *Rosa*$^{NICD1/+}$; *Rosa*$^{ZsG}$ (PNR) mice[23]. PNR mice developed frequent primary tumors in the liver but rare metastases. We then reported the generation of multiple cancer organoid lines derived from the PNR tumors. PNR organoid lines varied in their in vivo tumorigenicity with a subset being tumorigenic and metastatic upon orthotopic transplantation and the others generating no tumors in vivo (Fig. 1a)[8]. To pinpoint the molecular mechanisms driving metastasis in the PNR models, we performed a comparative RNAseq analysis of metastatic and nonmetastatic PNR tumor and organoid samples to identify genes associated with metastasis. The same analysis was done for a hepatocellular carcinoma (HCC) mouse model, PPTR (*Prominin1*$^{CreERT2}$; *Pten*$^{flx/flx}$; *Tp53*$^{flx/flx}$; *Rosa*$^{ZsG}$) mice, and combined to identify genes commonly associated with liver cancer metastasis (Supplementary Fig. 1a)[8,23]. The RNAseq data have been published previously (GSE94583). Among the top upregulated genes in metastatic PNR and PPTR tumors and organoids was *Rrm2* (Fig. 1b and Supplementary Fig. 1a), the catalytic M2 subunit of the ribonucleotide reductase (RNR) complex. RNR catalyzes the formation of deoxy-ribonucleoside diphosphate (dNDP) from

NDP. We found the expression of the other two RNR subunits, *Rrm1* and *Rrm2B*, did not show consistent association with the metastatic potential of tumors and organoids from both models (Fig. 1b and Supplementary Fig. 1b). Analysis of a previously published gene expression profiles of 88 HB patient tumors[24] also showed a significant association of *Ki67* and *RRM2* expression, not *RRM2B*, with both the pathologically- and molecularly-defined HB risk groups (Fig. 1c).

Since Rrm2 and Rrm2B were the catalytic subunits of the RNR complex, we did a more detailed validation of these two RNR M2 subunits in HB mouse and patient tumors. Due to its highly homologous but shorter protein sequence compared to Rrm2, no Rrm2B-specific antibodies were available for immunohistochemistry (IHC) validation. IHC on PNR tumors using an antibody recognizing both Rrm2 and Rrm2B proteins showed an increase of the number of (Rrm2 + Rrm2B)$^+$ cells in the metastatic tumors compared to the primary tumors (Supplementary Fig. 2). To accurately compare their expression in HB patient tumors, we developed RNAscope in situ hybridization assays specific to human *RRM2* and *RRM2B* mRNAs and showed that *RRM2* was expressed at a much higher level than *RRM2B* in all three HB PDXs we examined (Fig. 1d, iv–vi vs. vii–ix). *RRM2* positivity in HB PDX tumors was well aligned with that of a cell proliferation marker Ki67 (Fig. 1d, iv–vi vs. x–xii), supporting an association between *RRM2* expression and HB tumor malignancy.

### RRM2 and RRM2B are involved in distinct cellular processes in HB patient tumors.

Because of the difference association of RRM2 and RRM2B with HB patient prognosis, we performed a series of transcriptomic analysis of HB patient tumors with an effort to understand the systemic involvement of RRM2 and RRM2B in HB tumorigenesis. Using the same set of the previously published HB patient tumor microarray dataset in Fig. 1c (GSE75271)[24], we identified a set of RRM2 and RRM2B hub genes whose expression was highly correlated with *RRM2* and *RRM2B* expression, respectively, including both the upstream regulators and downstream targets (Fig. 2a). No overlap was found between the RRM2 and RRM2B hub genes (Fig. 2b and Supplementary Data 1). Because of the relatively small sample size of the HB patient tumors, RNA-seq data of 374 HCC patient tumors in The Cancer Genome Atlas (TCGA) database[25] were also analyzed as RRM2 was also associated with tumor progression in our PPTR HCC mouse model (Supplementary Fig. 1b). Similarly, RRM2 and RRM2B hub genes were identified in HCC patient tumors and found no overlap (Fig. 2c, d and Supplementary Data 1). Interestingly, limited overlap was found between the two tumor types for their RRM2 or RRM2B hub genes (Supplementary Fig. 3). However, a hypergeometric distribution method found that RRM2 hub genes in HB and HCC tumors are similarly enriched in the pathways involved in cell proliferation and DNA repair (RRM2-associated pathways, or PTWAY$^{RRM2}$) while RRM2B hub genes in both tumor types showed participation in stress and inflammatory response pathways (RRM2B-associated pathways, or PTWAY$^{RRM2B}$) (Fig. 2e). We then ranked the HB and HCC patient tumors based on their *RRM2* or *RRM2B* expression and selected tumors ranked top 1/3 and bottom 1/3 for each gene (*RRM2*$^{high}$ and *RRM2B*$^{high}$ tumors, respectively) to examine the enrichment of the top ten pathways of PTWAY$^{RRM2}$ and PTWAY$^{RRM2B}$ in these tumors. We found that in both cancer types, PTWAY$^{RRM2}$ were consistently and significantly enriched in *RRM2*$^{high}$ tumors, so were PTWAY$^{RRM2B}$ in *RRM2B*$^{high}$ tumors (Fig. 2e). It was noted that the activities of PTWAY$^{RRM2B}$ were generally low in *RRM2*$^{high}$ HB and HCC tumors and vice versa (Fig. 2e). These results indicate that RRM2 and RRM2B are involved in distinct cellular

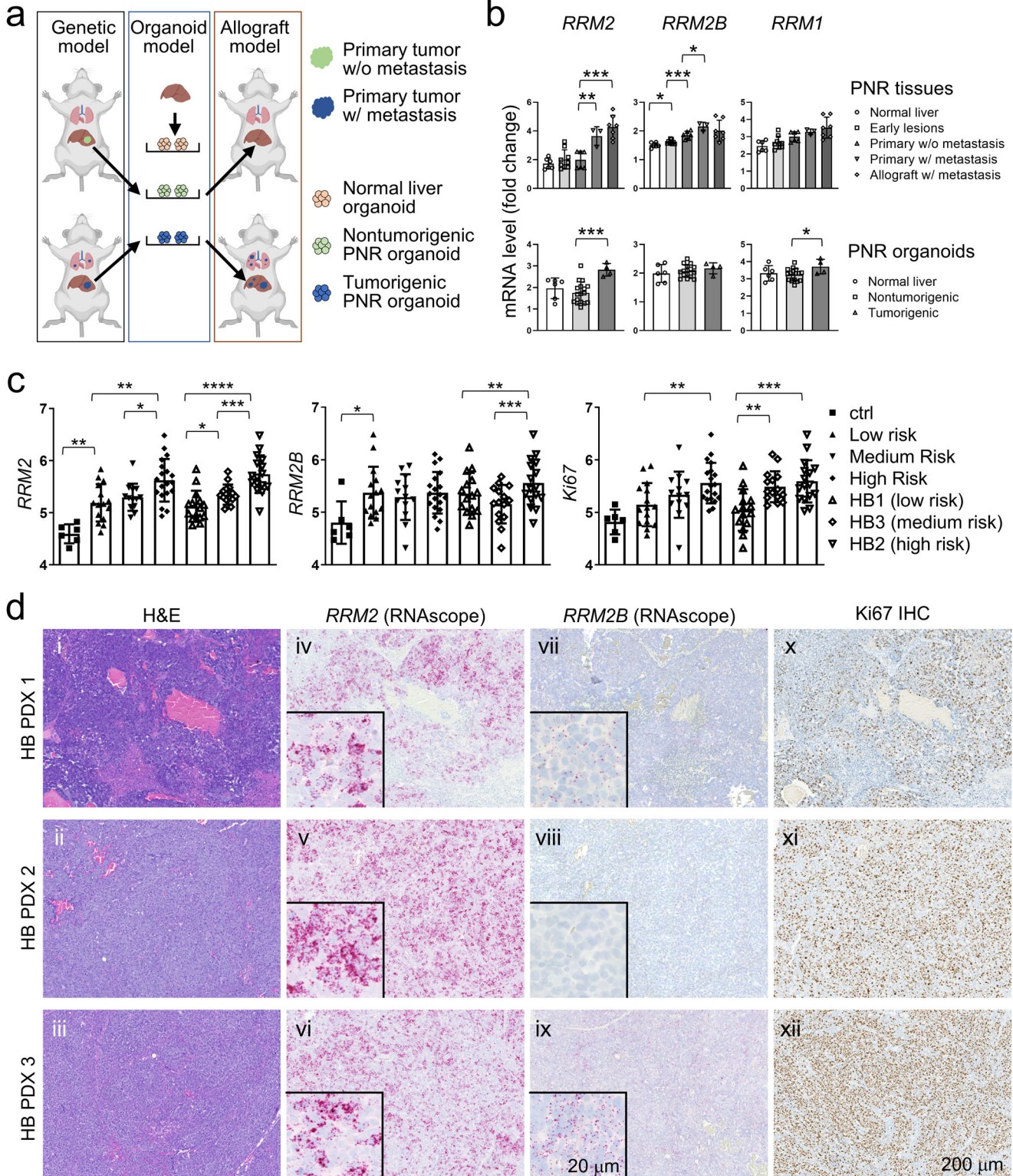

**Fig. 1 RRM2, not RRM2B, is associated with HB progression in mice and patients. a** Schematic illustration (created with BioRender.com) showing the establishment of PNR genetic mouse model, organoid model, and orthotopic allograft models. **b** Quantitative comparison of the gene expression of the three RNR subunits in PNR tumor tissues ($n = 6, 9, 6, 3,$ and 7 for the five indicated groups, respectively) and organoids ($n = 6, 17,$ and 4 for the three indicated groups, respectively). **c** Expression of the three RNR subunits in the control liver ($n = 6$) and pathologically-defined HB patient risk groups (low, medium, and high; $n = 15, 13,$ and 19, respectively) and molecularly-defined risk groups (HB1, HB3, and HB2 from low- to high-risk; $n = 15, 15,$ and 16, respectively) from a publicly available HB transcriptomics database[24]. **d** H&E staining (i–iii), RNAscope staining for *RRM2* (iv–vi) and *RRM2B* (vii–ix), and Ki67 IHC staining (x–xii) on the serial sections of three HB PDX tumors. All images share the same 200 μm scale bar in (xii). Insets in (iv–ix): higher magnification images of the corresponding RNAscope staining and share the 20 μm scale bar in the inset (ix). Student *t* test: *P* values: * <0.05; ** <0.01; *** <0.001; **** <0.0001.

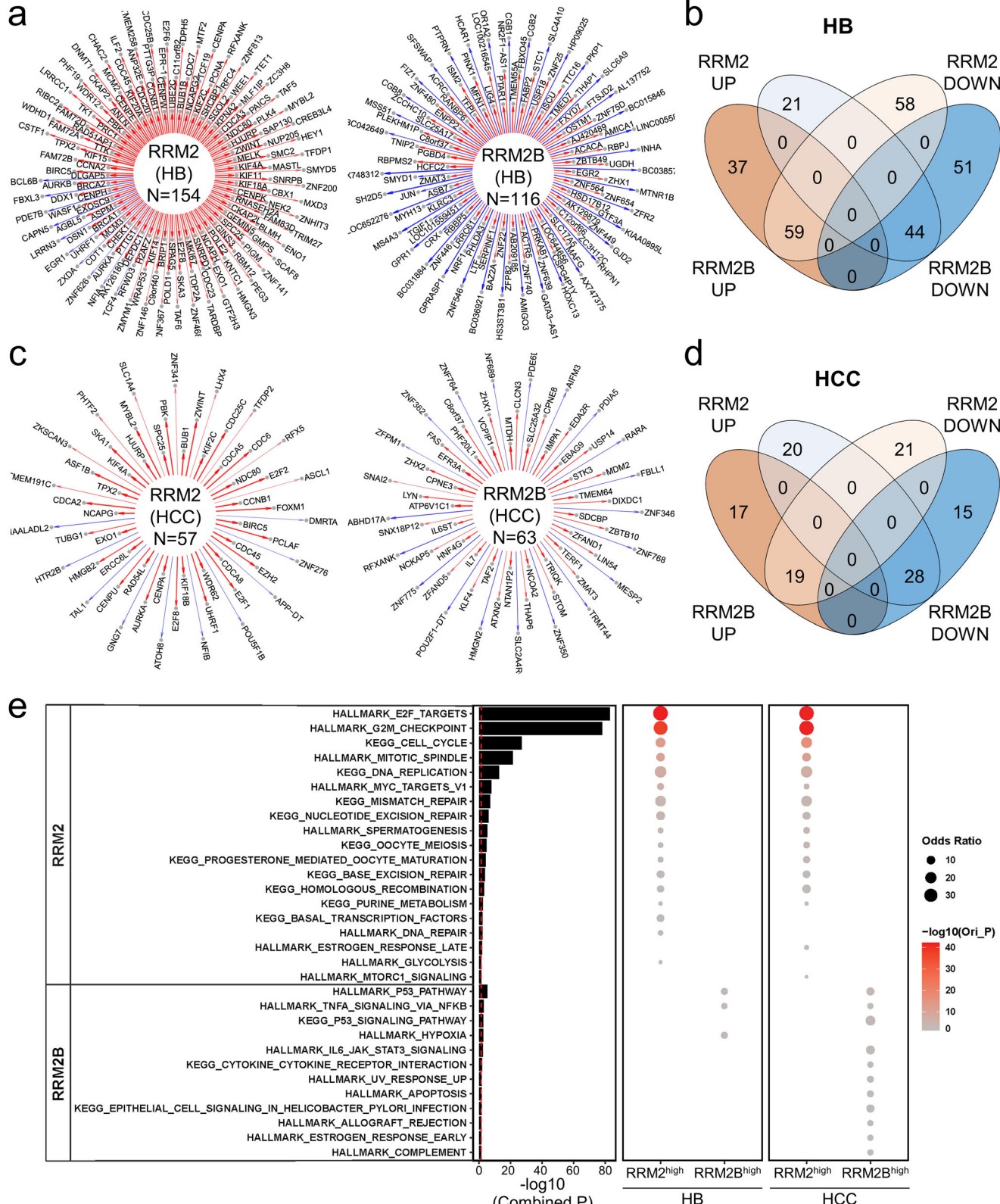

**Fig. 2 NetBID identified distinct hub genes and associated signaling pathways of RRM2 and RRM2B in HB and HCC patient tumors. a** The RRM2 and RRM2B networks in 88 HB patient tumors inferred by SJARACNe. Edge width corresponds to the correlation strength measured by mutual information. Red and blue edges indicate positive and negative correlations between the expression of *RRM2* or *RRM2B* and their individual hub genes. **b** Venn plot showing the overlap of the four indicated *RRM2* and *RRM2B* hub gene lists identified in HB patient tumors. **c** The RRM2 and RRM2B networks in 374 HCC patient tumors similarly defined as in (**a**). **d** Venn plot showing the overlap of the four indicated *RRM2* and *RRM2B* hub gene lists identified in HCC patient tumors. **e** Gene set enrichment analysis of HALLMARK and KEGG gene sets of the 4 hub gene lists using Fisher's Exact Test. The size and color intensity indicate the odds ratio and statistical significance, respectively. The *P* values of the bar plot were combined from HB and HCC primary patient cohorts using the Stouffer method.

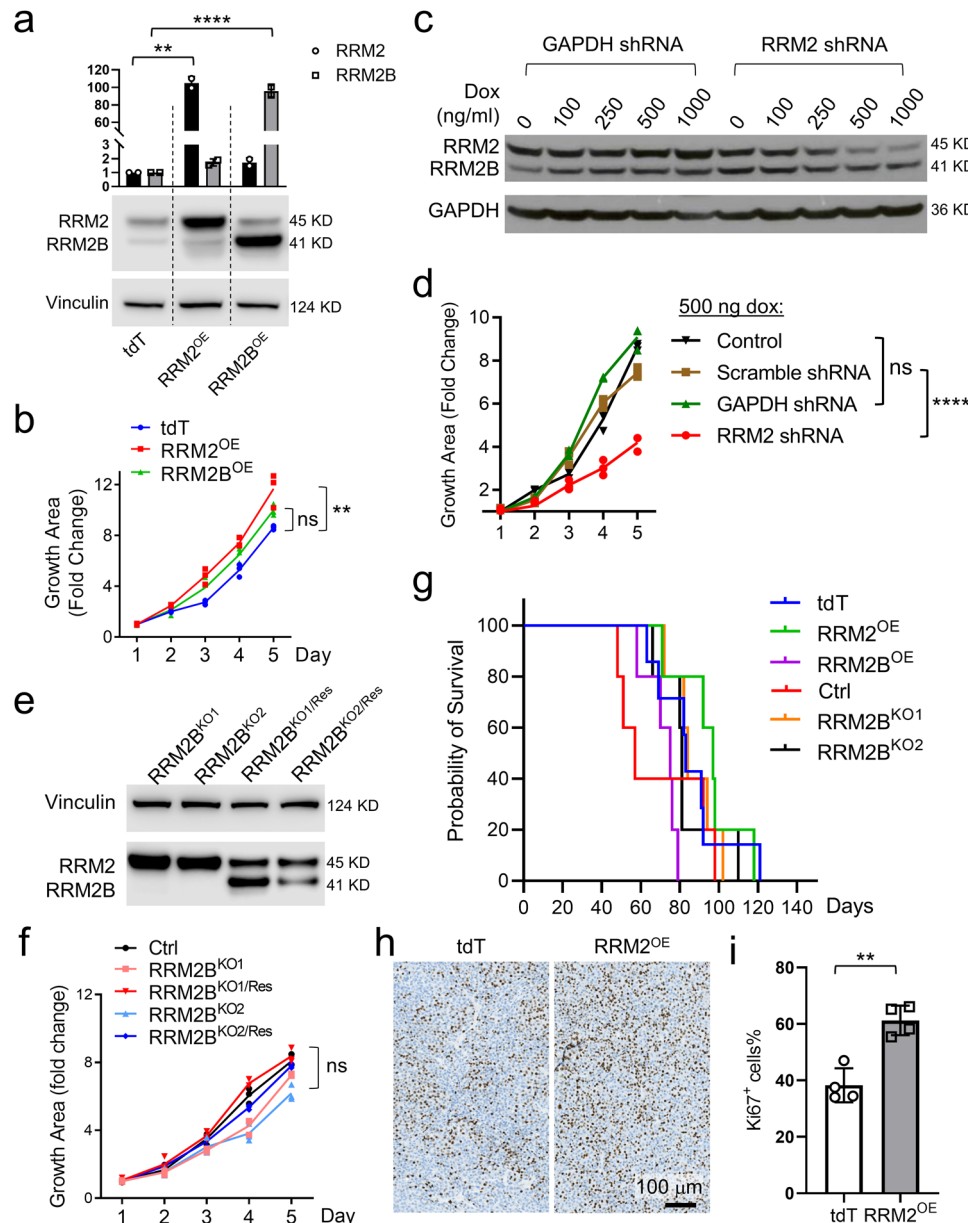

**Fig. 3 RRM2, not RRM2B, supports HB cell growth in vitro and in vivo. a** Quantitative RT-PCR and immunoblotting of RRM2 and RRM2B in the indicated HepG2 cells overexpressing a *tdT* control vector, *RRM2*, or *RRM2B*. **b** Growth curves of *tdT*, *RRM2^OE^*, and *RRM2B^OE^* HepG2 cells ($n = 3$ for each group). **c** RRM2 and RRM2B immunoblotting showing a dose-dependent knockdown of *RRM2* in HepG2 cells expressing a doxycycline (dox)-inducible *RRM2* shRNA. *GAPDH* shRNA was used as a control. **d** Growth curves of the indicated HepG2 cells treated with 500 ng dox ($n = 3$ for each group). **e** RRM2 and RRM2B immunoblotting in *RRM2B^KO^* and *RRM2B^KO/Res^* HepG2 cells. **f** Growth curves of the indicated *RRM2B*-manipulated HepG2 cells ($n = 3$ for each group). **g** Survival curves of the NSG mice orthotopically transplanted with the indicated *RRM2-* and *RRM2B*-manipulated HepG2 cells ($n = 7$ for the tdT group; $n = 5$ for each of the other groups). No significant statistical differences were observed between any groups. **h** Representative Ki67 IHC images of the *tdT* and *RRM2^OE^* HepG2 allograft tumors. Both images share the same 100 μm scale bar. **i** Quantification of the percentage of Ki67+ cells in the most proliferative regions in the *tdT* and *RRM2^OE^* HepG2 allograft tumors. Ki67+ cells in four tumors from each group were counted. Student $t$ test: P values: ** <0.01.

processes in both HB and HCC patient tumors. Following these leads, we performed more experiments to determine the differential involvement of RRM2 and RRM2B in HB cell proliferation and stress response.

**RRM2, not RRM2B, is associated with HB cell proliferation.** To confirm the different involvement of RRM2 and RRM2B in HB cell proliferation, we overexpressed *RRM2* and *RRM2B* in a human HB cell line HepG2[26,27] (Fig. 3a). RRM2 has been shown to have higher RNR activity than RRM2B[12] and we confirmed the overall higher levels of free nucleoside diphosphate (NDP), deoxy-

NDP (dNDP) and deoxynucleoside triphosphate (dNTP) in *RRM2*-overexpressing (*RRM2^OE^*) cells than *RRM2B^OE^* cells via targeted liquid chromatography/mass spectrometry (Supplementary Fig. 4). Accordingly, *RRM2^OE^* cells showed a modest but significant increase in their growth rate than control cells transfected with a tdTomato (tdT) plasmid while changes in *RRM2B^OE^* cell growth did not reach statistical significance (Fig. 3b). The growth-promoting role of RRM2 in HB cells was further supported by a doxycycline-inducible shRNA *RRM2* knockdown in HepG2 cells which led to a significant reduction in cell growth (Fig. 3c, d). DepMap Portal shows *RRM2* is an essential gene and

its knockout leads to the death of all cancer cell lines tested. Our attempt to knockout *RRM2* in *RRM2B^OE* HepG2 cells via CRISPR/Cas9 also failed. We noticed an unusual bias towards in-frame indels that persisted in culture (Supplementary Fig. 5). A wildtype copy of *RRM2* was also retained by all cells survived long-term cultivation. *RRM2B* is not an essential gene and we successfully generated two *RRM2B* knockout single-cell clones (*RRM2B^KO1* and *RRM2B^KO2*) of HepG2 cells via CRISPR/Cas9 (Fig. 3e). *RRM2B*-rescued cells were also generated for both clones by re-expressing *RRM2B* (*RRM2B^KO1/Res* and *RRM2B^KO2/Res* cells) (Fig. 3e). No growth differences were found in these *RRM2B*-manipulated cells compared to control wildtype HepG2 cells, indicating that *RRM2B* is not involved in HB cell growth (Fig. 3f). When the control, *RRM2^OE*, *RRM2B^OE*, *RRM2B^KO1* and *RRM2B^KO2* cells were orthotopically transplanted into the NOD scid gamma (NSG) mouse liver, no significant differences were found in the animal survival between different groups (Fig. 3g). It was not very surprising that *RRM2* overexpression did not lead to significantly shortened animal survival considering the moderate effect of *RRM2* overexpression on HepG2 cell growth in vitro (Fig. 3b). When examining tumor proliferation via Ki67 IHC, we noticed that the cell proliferation was highly heterogenous in HepG2 tumors in vivo, which was likely being affected by many environmental factors. However, we found that the most proliferative regions of the *RRM2^OE* tumors had a significantly higher percentage of Ki67+ cells than those of the tdT tumors (Fig. 3h, i). Overall, these results indicate that RRM2 is the dominant RNR M2 subunit supporting HB cell proliferation. While highly homologous in protein sequences to RRM2, RRM2B is not involved in HB cell growth.

**Drug treatment suppresses RRM2 but induce RRM2B in HB cells in vitro.** To examine the involvement of RRM2 and RRM2B in HB stress response, we decided to test their response to drug treatment. We tested an RRM2 inhibitor, triapine[28] and a WEE1 inhibitor MK1775 which has been shown to inhibit cell cycle via depleting RRM2[29], in HB cells in comparison to other standard liver cancer chemotherapies including cisplatin, gemcitabine, vincristine, and SN38, an active form of irinotecan[30]. Due to the limited cell resources for HB, we had only two HB cell lines for drug testing, HepG2 and HB214. HB214 cells were derived from a previously reported HB PDX model[31], and they generated tumors with faithful HB histopathology when transplanted into NSG mouse liver[32]. Both cell lines were fairly resistant to all the drugs tested with high IC$_{50}$ values (Supplementary Table 1), and triapine and MK1775 did not show higher efficacies compared to the other drugs (Fig. 4a). Consistent with their RRM2-inhibiting function, triapine and MK1775 treatment led to a significant reduction in the nucleotide levels in HepG2 cells along with a strong induction of the cell cycle suppressor p21 and the DNA damage marker γ-H2AX (Supplementary Fig. 6). When examining the response of the two RNR M2 subunits to drug treatment, we noticed that all the drugs tested led to a significant reduction in the RRM2 protein levels in both cell lines. However, interestingly, they also led to a marked increase in RRM2B protein levels (Fig. 4b). RRM2B upregulation has not been reported in the context of RRM2 inhibition or to chemotherapy. RNAseq transcriptomic profiling of cisplatin-treated HepG2 cells showed that such drug-induced changes in RRM2 and RRM2B happened at the mRNA level as well (Supplementary Fig. 7a). A more detailed time course study on cisplatin-treated HepG2 cells revealed an upregulation of RRM2 at first within the first 24 h, before its level started decreasing accompanied by an increase in RRM2B levels which occurred around 18–24 h and became stable after 48 h (Supplementary Fig. 7b). No drug-induced changes in

RRM1 protein levels were observed except for a slight reduction at the highest drug concentrations (Supplementary Fig. 7c). RRM1 immunoprecipitation using MK1775- and gemcitabine-treated HepG2 cells showed an evident RRM1-RRM2B binding dominant over RRM1-RRM2 binding (Supplementary Fig. 8), suggesting an increase in RRM1-RRM2B complex upon drug treatment. RRM2B has been shown to be regulated by the tumor suppressor p53 (encoded by the *TP53* gene)[33]. HepG2 cells have a wildtype *TP53* gene[34]. When examining HepG2 cells treated with cisplatin, triapine and gemcitabine, we found that there was a well correlated increase in p53 and RRM2B protein levels with increasing drug concentrations (Supplementary Fig. 9a). Two HCC cell lines with *TP53* mutations, PLC/PRF/5 and Hep3B[35], when treated with gemcitabine and a common HCC drug sorafenib, failed to induce RRM2B (Supplementary Fig. 9b). Gemcitabine was not very effective towards both HCC cell lines (Supplementary Fig. 9c) which may be explained by its strong induction of RRM2 in the HCC cells (Supplementary Fig. 9b). These results suggested a differential RRM2 and RRM2B regulation in HB and HCC cells in response to drug treatment potentially in a p53-dependent manner.

**Both RRM2 and RRM2B contribute to HB cell drug resistance, but only RRM2B promotes the post-treatment recovery of HB cells.** To determine the involvement of RRM2 and RRM2B in HB cell drug response, we treated tdT, *RRM2^OE* and *RRM2B^OE* HepG2 cells with six common liver cancer drugs and the two RRM2 inhibitors triapine and MK1775. We found overexpression of *RRM2* resulted in only a mild but statistically significant increase in the IC50 of all the drugs tested except for SN38 (Fig. 5a, b). *RRM2B^OE* HepG2 cells showed a similar mild but statistically significant increase in drug resistance to an extent less than the *RRM2^OE* cells. Compared to the tdT cells, *RRM2B^OE* cells showed no differences in response to the two RRM2 inhibitors in addition to SN38 (Fig. 5a, b), supporting the specificity of these two inhibitors to RRM2. *RRM2B*-manipulated cells, including both *RRM2B^KO* clones and their *RRM2B*-rescued cells, were also tested for the same drugs. While no decrease in drug resistance was observed in either *RRM2B^KO* clones, re-expressing *RRM2B* led to a mild but significant increase in the IC$_{50}$ values of all the drugs except for SN38 (Fig. 5c, d, Supplementary Fig. 10, and Supplementary Table 1).

Since we have shown that increase in RRM2B by drug treatment did not stabilize until 48 h later (Supplementary Fig. 7b), we suspected that RRM2B might function mostly to improve the fitness of HB cells that had survived drug treatment. To test this hypothesis, we treated all the RRM2- and *RRM2B*-manipulated cells with cisplatin at different concentrations for three days. We confirmed that there was no drug-induced *RRM2B* upregulation in either *RRM2B^KO* clones (Supplementary Fig. 11), further confirming a complete *RRM2B* depletion in these cells. We then collected the surviving cells, seeded them at a low density, and monitored their recovery over 12 days. We found post-cisplatin treatment recovery was markedly affected by *RRM2B* manipulation but not that of *RRM2*. No difference in the number of colonies grown from the cisplatin-treated *RRM2^OE* and tdT control cells (Fig. 6a, b). But cisplatin-treated *RRM2B^OE* cells grew significantly more colonies compared to the control while both had significantly fewer colonies than their control (Fig. 6a, c). *RRM2B* re-expression partially rescued this phenotype, however, only in the cells treated at the lower concentration of cisplatin (Fig. 6a, c). To confirm the importance of RRM2B in the post-treatment recovery of HB cells, we repeated *RRM2B* knockout in HB214 cells (Supplementary Fig. 12a) and found *RRM2B* loss caused a consistent and

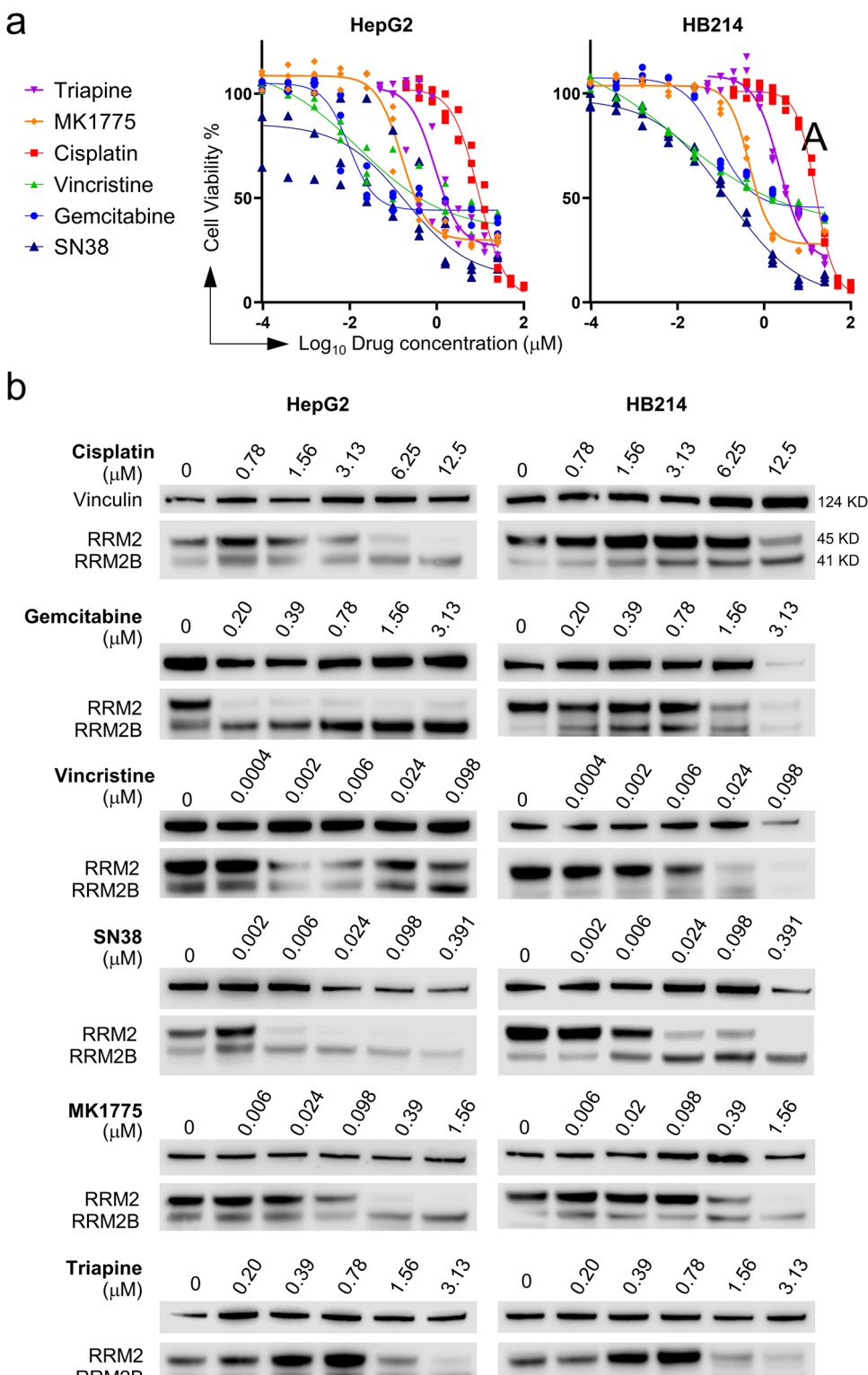

**Fig. 4 A drug-induced RRM2 to RRM2B subunit switch in HB cells in vitro. a** The dose-response curves of HepG2 and HB214 cells treated with the indicated chemotherapeutic agents and RRM2 inhibitors ($n = 3$ for each group). **b** RRM2 and RRM2B immunoblotting HepG2 and HB214 cells with the indicated drugs.

significant reduction in their post-cisplatin recovery (Supplementary Fig. 12b, c). These results, together, suggest a critical role of RRM2B, not RRM2, in supporting the post-treatment recovery of HB cells.

Next, we performed RNAseq of the *RRM2-* and *RRM2B-* manipulated HepG2 cells with or without cisplatin treatment to determine their transcriptomic changes in comparison to the PTWAY^RRM2 and PTWAY^RRM2B we previously identified in HB and HCC patient tumors (Fig. 2e). A comprehensive quality assessment revealed high mapping rates of the reads, high integrity of the libraries and high accuracy of gene expression quantification (Supplementary Fig. 13). Consistently with our observations of the

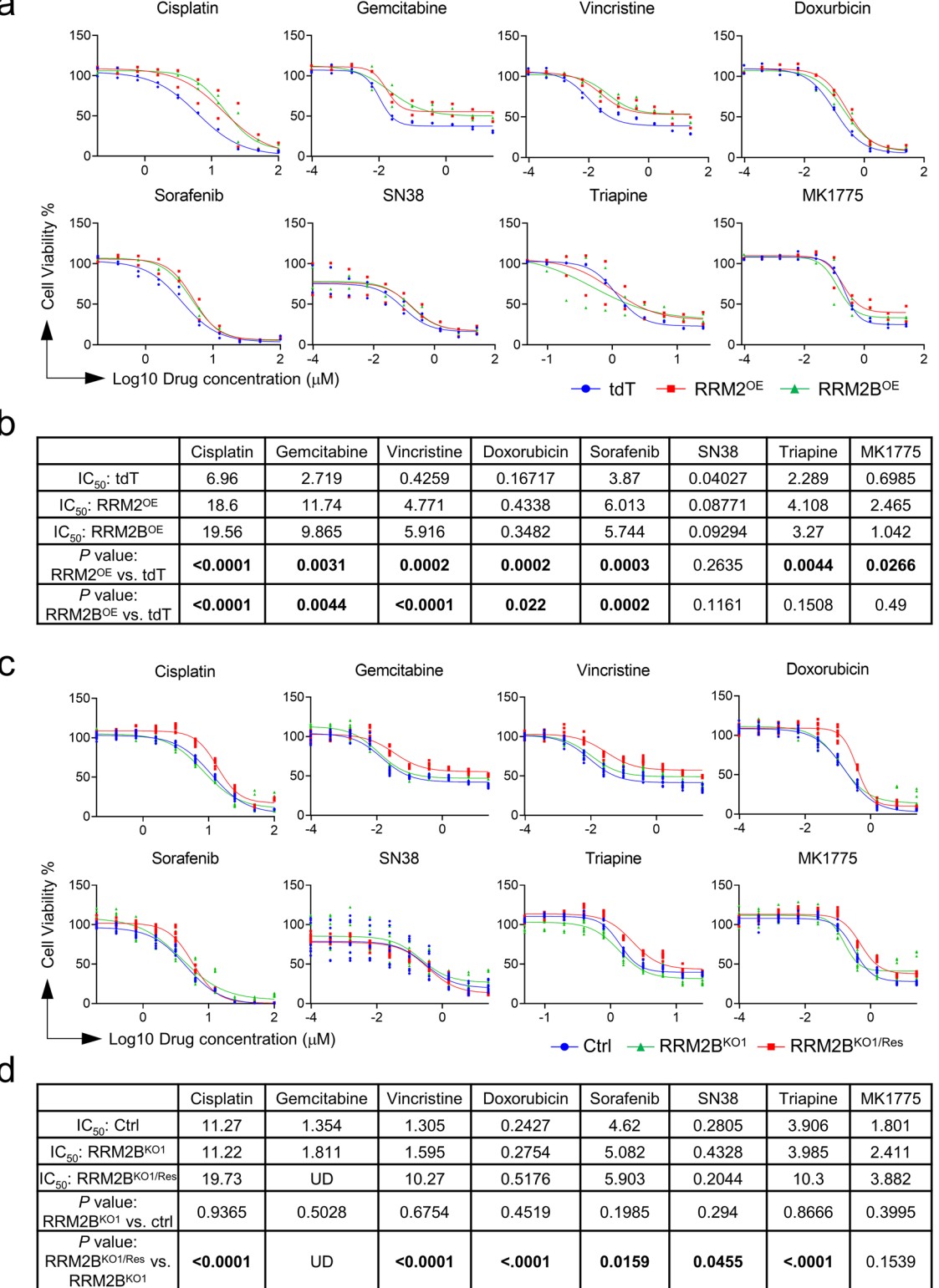

**Fig. 5 Both RRM2 and RRM2B contribute moderately to HepG2 cell drug resistance. a** The dose-response curves of *tdT*, *RRM2^OE* and *RRM2B^OE* HepG2 cells to the indicated drugs (*n* = 3 for each group). **b** List of the drug IC50 values and their comparisons between *tdT*, *RRM2^OE* and *RRM2B^OE* HepG2 cells. **c** The dose-response curves of control (wildtype), *RRM2B^KO1* and *RRM2B ^KO1/Res* HepG2 cells to the indicated drugs (*n* = 3 for each group). **d** List of the drug IC50 values and their comparisons between control (wildtype), *RRM2B^KO1* and *RRM2B ^KO1/Res* HepG2 cells. Extra Sum of Square F test was performed for (**b**, **d**). Statistically significant *P* values (<0.05) were indicated by bold font.

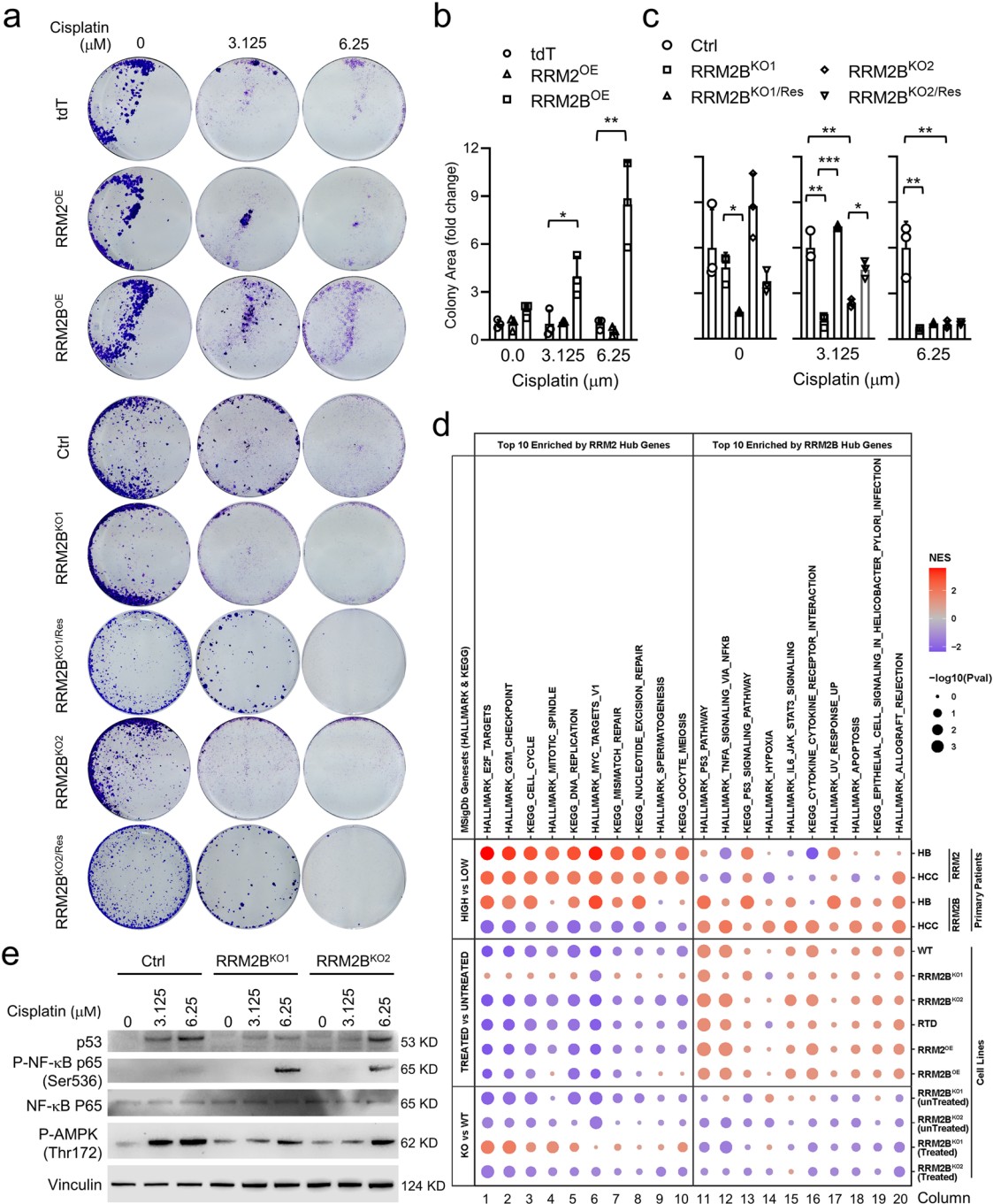

**Fig. 6 RRM2B, not RRM2, supports post-drug treatment recovery of HB cells in vitro. a** 12-day colony formation assay of the indicated *RRM2-* and *RRM2B*-manipulated HepG2 cells post cisplatin treatment at two different concentrations. **b, c** Quantitative comparison the colony area occupied by cells in (A) (*n* = 3 for each group). Student *t* test. *P* values: * <0.05, ** <0.01, *** <0.001. **d** Gene set enrichment analysis of HALLMARK and KEGG gene sets using pre-ranked gene list by fold change of gene expression across different comparisons of the indicated HB cells and patient tumors. The size and color intensity of the bubbles indicate the normalized enrichment score and statistical significance, respectively. **e** Immunoblotting of the indicated proteins in the cisplatin-treated *RRM2B^KO* cells.

drug-induced RRM2-to-RRM2B switching, we found that cisplatin treatment led to the activation of PTWAY^RRM2B and suppression of PTWAY^RRM2 in all treated cells compared to their untreated counterpart (Fig. 6d). When comparing untreated *RRM2B^KO* cells, cisplatin-treated *RRM2B^KO* cells also showed a similar activation of PTWAY^RRM2B. However, compared to the control cells with no *RRM2B* loss, *RRM2B^KO* cells had a consistent and negative enrichment of PTWAY^RRM2B either with or without cisplatin treatment. These results suggest that RRM2B participated in but

did not dictate the drug-induced activation of PTWAY^RRM2B, which was composed predominantly of stress response pathways such as p53 and NFκB pathways (Supplementary Fig. 14). To validate the participation of RRM2B in the stress response pathways, we performed immunoblotting to detect p53, phospho-NFκB (Ser536), and phospho-AMPK (Thr172) which is another important stress response pathway[36], in the two *RRM2B^KO* clones with and without cisplatin treatment. We found that the levels of p53 induction and AMPK phosphorylation by the drug treatment

were indeed much lower in the $RRM2B^{KO}$ cells (Fig. 6e). However, interestingly, NFκB showed a significantly higher level of phosphorylation in both $RRM2B^{KO}$ clones, particularly in the cells treated with the higher concentration of cisplatin (Fig. 6e). Compared to the negative enrichment of the NFκB pathway in cisplatin-treated $RRM2B^{KO}$ cells compared to control indicted by RNAseq profiling, this unexpected augmentation of NFκB phosphorylation suggests a complex stress response regulation at both transcriptional as well as post-transcriptional level. Nevertheless, our data clearly demonstrated the participation of RRM2B in the stress response of HB cells and its importance in supporting their post-drug treatment recovery.

**A reversed RRM2B to RRM2 subunit switching in HB cells recovering from drug treatment.** Our data suggested that inhibiting RRM2B would work as an effective combinatorial treatment to standard chemotherapies. However, no RRM2B-specific inhibitors have been developed likely due to the low expression of this RNR M2 subunit in growing tumor cells. We tested two drugs that could potentially inhibit RRM2B, deferoxamine[37] and KU60019[38], however, neither of which showed RRM2B-specific inhibition (Supplementary Fig. 15). Since our data indicated that RRM2 was important to HB cell growth, we suspected that HB cells would regain $RRM2$ expression in order to support their regrowth after treatment. In turn, RRM2 inhibition would be an effective approach to prevent HB relapse. Indeed, when cisplatin-treated HepG2 cells were allowed to regrow, we found that the level of RRM2 did come back up and became the dominant RNR M2 subunit again while RRM2B levels decreased (Fig. 7a). This reversed RRM2B-to-RRM2 switching occurred quickly in cells that were initially treated with a relatively low concentration of cis (3.1 μM), and much more slowly and at a markedly lower level in cells treated with a high dose of cis (12.5 μM) (Fig. 7a), consistent with their slow post-treatment growth observed previously (Fig. 6a).

Based on these results, we tested the efficacy of combining RRM2 inhibitors with chemotherapy in preventing HB relapse. To identify the synergistic combinations, we first tested HepG2 and HB214 cells for combinatorial treatments of the two RRM2 inhibitors, triapine and MK1775, with other chemotherapeutic agents in vitro. We found that SN38, the active form of irinotecan, showed good synergy with both triapine and MK1775 in HB214 among all the combinations we tested (Fig. 7b and Supplementary Table 2). We then tested triapine/irinotecan and MK1775/irinotecan combinatorial treatment in the HB214 subcutaneous PDX model. Triapine treatment caused rapid body weight loss and all animals had to be removed from the study early. Treatment with irinotecan and a low dose of MK1775, which had minimal toxicity or anti-tumor efficacy by itself, showed no additional benefit in tumor suppression compared to irinotecan alone during the three weeks of treatment. However, a modest but statistically significant delay in tumor relapse was observed in mice which had previously received the MK1775/irinotecan combinatorial treatment (Fig. 7c). Based on this result, we further tested the possibility of continuing this low dose MK1775 as a "maintenance therapy" in a second in vivo test. We shortened MK1775/irinotecan combinatorial treatment to two weeks and then MK1775 alone afterwards. We found that, compared to irinotecan monotherapy, this MK1775/irinotecan-MK1775 treatment regimen was able to delay tumor relapse for approximately 10 days (Fig. 7d).

To determine if the RNR M2 subunit switching we observed in vitro occurred in vivo, we collected HB214 tumors two days and 34 days post irinotecan treatment and performed $RRM2$ and $RRM2B$ RNAscope staining on tumor sections. We confirmed

that, indeed, there was an evident RRM2-to-RRM2B switching in tumors collected two days after irinotecan treatment. Compared to the untreated HB214 tumors, the treated tumors showed a marked loss of the typical punctate signals of $RRM2$ mRNA in many areas as well as the loss of the tight association between $RRM2$ and cell proliferation indicated by Ki67 IHC while $RRM2B$ signal was markedly increased (Fig. 7e). A reversed RRM2B-to-RRM2 switching was evident in the relapsed tumor 34 days post treatment with the regaining of $RRM2$ punctate signals and its association with Ki67 positivity when $RRM2B$ dropped (Fig. 7e). These observations were also confirmed by RRM2/RRM2B immunoblotting using tumors collected at different time points post drug treatment (Fig. 7f). Together, these results validated the drug- and relapse-associated RNR M2 subunit switching in vivo and demonstrated the promise of combining RRM2 inhibition with chemotherapy to prevent HB relapse.

**RRM2B level is associated with drug response in HB patient tumors.** Lastly, we examined whether our findings on RNR M2 subunit switching in HB cells and PDXs also applied to primary HB patient tumor samples. Because HB is very rare, and nearly all children with HB receive neoadjuvant chemotherapy prior to tumor resection, we were unable to obtain matched pre- and post-treatment patient tumor samples to directly determine treatment-induced changes in $RRM2$ and $RRM2B$ expression. Instead, we identified three Stage I HB patient tumors, within which patient tumor 1 (PT1) had received less neoadjuvant treatment than PT2 and PT3. We performed $RRM2$ and $RRM2B$ RNAscope staining and found tumor cells in PT1 expressed predominantly $RRM2$ with no evident $RRM2B$ expression (Fig. 8a, iii vs. iv and Supplementary Fig. 16). In PT2 and PT3 tumors, $RRM2$ staining was mostly a diffused hue instead of the standard punctate signals (Supplementary Fig. 16), similar to that of drug-treated HB214 PDX tumors (Fig. 7e). In the remaining viable tumor cells within these two tumors, we found much lower levels of $RRM2$ but higher levels of $RRM2B$ compared to PT1 (Fig. 8a, vii & xi vs. iii, viii & xi vs. iv). We also obtained two freshly resected HB patient tumors that had received standard chemotherapy and performed RRM2 and RRM2B immunoblotting. Both tumor samples showed a markedly higher level of RRM2B than RRM2 (Fig. 8b). Compared to their matched background liver samples, RRM2B levels in the tumors were also significantly higher while RRM2 was at a comparable low level (Fig. 8b). Although we were unable to obtain direct supporting evidence for chemotherapy-induced RRM2-to-RRM2B switch in primary patient tumors due to the lack of clinical resources, our data are consistent with the notion that RRM2B is the dominant RNR M2 subunit in chemotherapy-treated HB patient tumors.

Based on all the data we collected in HB cells, PDXs and primary patient tumors, we propose a dynamic RNR M2 subunit switching model in supporting HB cell proliferation versus survival—RRM2 is the dominant RNR M2 subunit supporting HB cell growth; RRM2 switches to RRM2B when drug treatment inhibits cell proliferation and induces stress; once helping cells survive the stress, RRM2B switches back to RRM2 for its better ability to support tumor cell regrowth (Fig. 8c).

**Discussion**
In this study, we showed that the two RNR M2 subunits RRM2 and RRM2B, despite sharing highly homologous protein sequences, are involved in distinct cellular processes and are regulated dynamically to support HB cell growth and stress response. Upregulation of RRM2 was tightly associated with HB prognosis and tumor cell proliferation. Treatment with standard chemotherapies and RRM2 inhibitors effectively suppressed

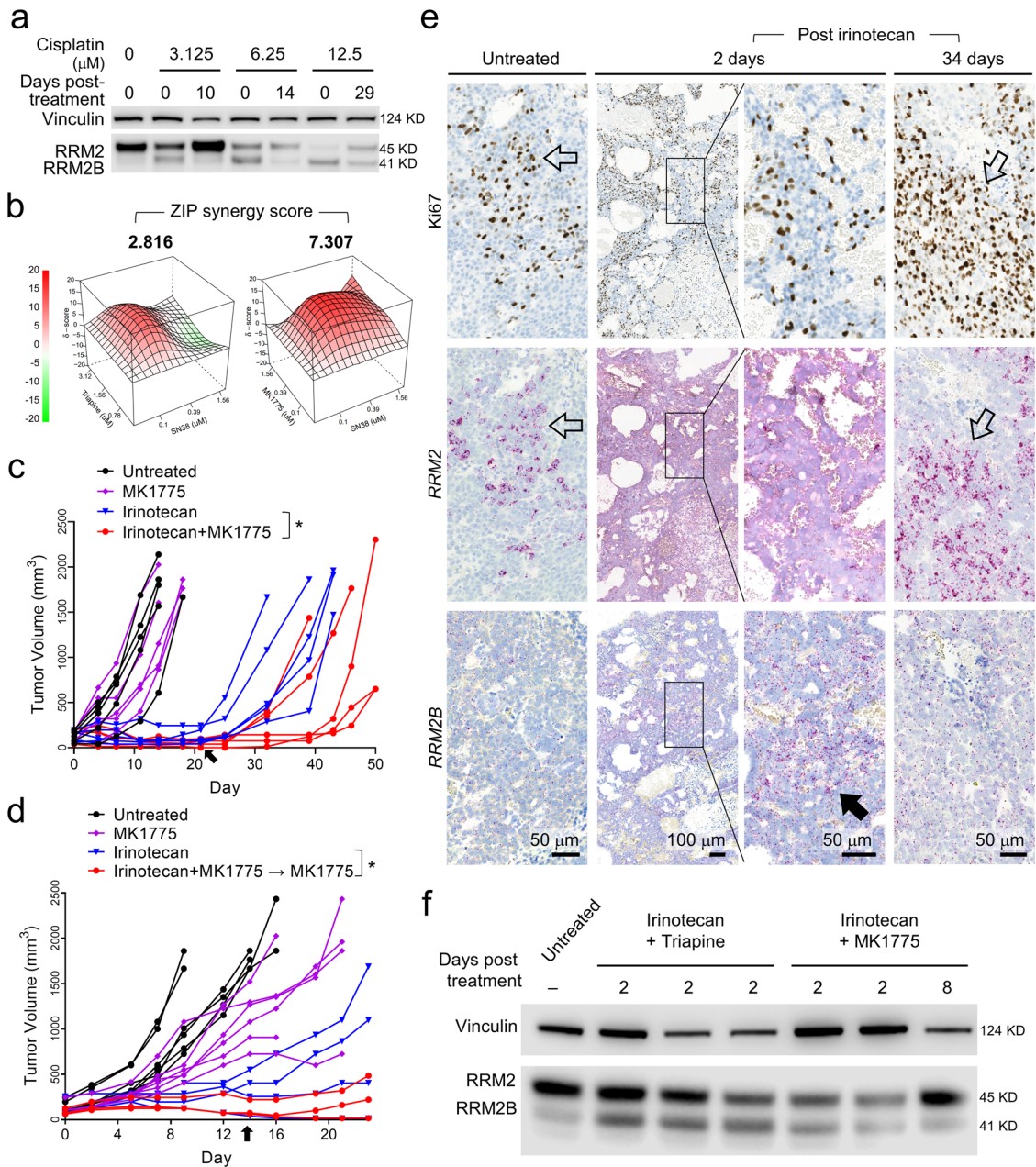

**Fig. 7 A reversed RRM2B-to-RRM2 subunit switching in HB cells during post-drug treatment regrowth can be targeted to delay HB tumor relapse in vivo. a** RRM2 and RRM2B immunoblotting using HepG2 cells recovered from the indicated cisplatin treatment. **b** ZIP synergy analysis of the two RRM2 inhibitors in combination with SN38 in HB214 cells. **c** Tumor volume measurements in mice treated with the indicated drug or drug combination. Arrow: drug withdraw on Day 21. **d** Tumor volume measurements in mice treated with the indicated drug or drug combination. Arrow: drug withdraw on Day 14. MK1775 was maintained in the last group (red line). **e** Ki67 IHC and *RRM2* and *RRM2B* RNAscope in untreated HB214 tumors and those two days and 34 days post irinotecan treatment. Open arrows: co-localization of Ki67+ and *RRM2*+ cells in untreated and relapsed HB214 tumors; solid arrow: transient increase in *RRM2B* expression in HB214 tumors two days post irinotecan treatment. Images on the same column share the same scale bar. **f** RRM2 and RRM2B immunoblotting of the HB214 tumors collected post the indicated treatment. Extra Sum of Square F test was performed for (**c**, **d**). *P* value, * <0.05.

RRM2 but induce the upregulation of RRM2B. We confirmed that RRM2B had a lower RNR enzymatic activity than RRM2 as reported and had limited participation in the proliferation or the initial drug response of HB cells. However, RRM2B was critical to the fitness of the HB cells that survived drug treatment, allowing them to relapse more efficiently post drug treatment. When HB cells resumed proliferation during relapse, RRM2 became the dominant RNR M2 subunit again while RRM2B gradually dropped to its low, pre-drug treatment level. Adding a low-dose of RRM2 inhibitor MK1775 to irinotecan was able to delay tumor

relapse in an HB PDX model compared to irinotecan treatment alone. Computational analysis of HB and HCC patient transcriptomic profiles and our RNR-manipulated HB cells revealed distinct cellular network associated with these two RNR M2 subunits, that PTWAY[RRM2] involves primarily in cell proliferation and DNA repair while PTWAY[RRM2B] participates heavily in stress and inflammatory responses. RRM2 is well known for its upregulation in adult solid tumors and its essential role in supporting cell proliferation[15–19]. RRM2B has been reported to contribute to stress response and drug resistance of

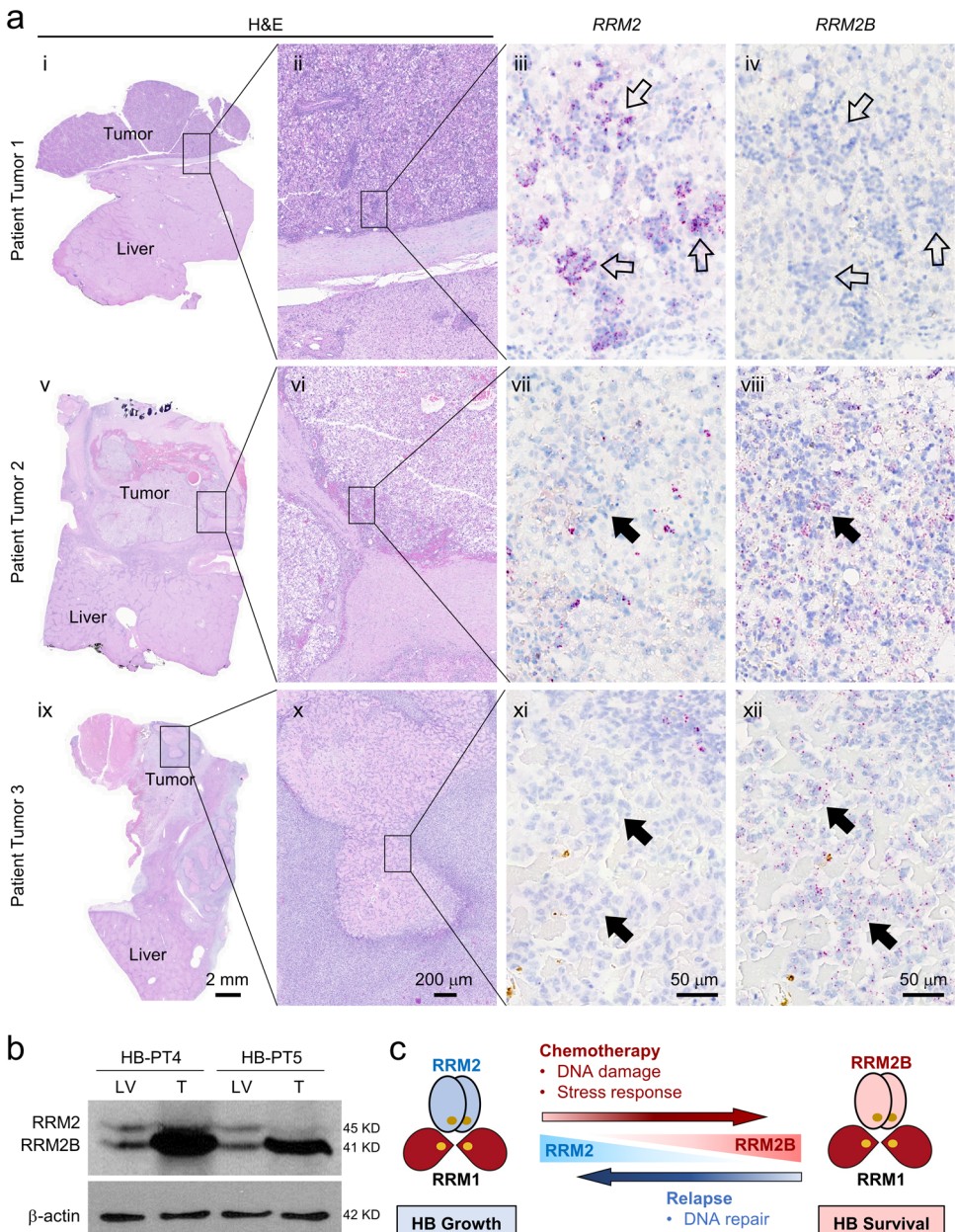

**Fig. 8 RRM2B level is associated with drug response in HB patient tumors. a** H&E staining, RNAscope staining for *RRM2* and *RRM2B* on serial sections of three post-chemotherapy primary HB patient tumors. Open arrows in (iii, iv): *RRM2*+/*RRM2B*− cells in viable tumor areas; arrows in (vii, viii) and (xi, xii): cells with evident *RRM2B* expression but not *RRM2* expression in the tumor areas stressed by chemotherapy. Images on the same column share the same scale bar. **b** RRM2 and RRM2B immunoblotting of two primary HB patient tumors collected post-chemotherapy. **c** A working model of RRM2 and RRM2B subunit switching in HB growth, drug response, and relapse.

cancer cells[39–41]. However, this study, to our knowledge, is one of first studies that has revealed a dynamic, reversible switching between the two RNR M2 subunits, and depicted a biological picture on how HB cells transit between a growing state and a surviving state from a point of RNR dynamics.

Rare pediatric cancers present one of the greatest challenges to the oncology community because of the limited patients for therapeutic and biological investigations. HB is a very rare cancer overall although the most common primary liver cancer in children. Our study is limited for the small number of HB cell lines tested. However, the consistent results we were able to obtain from the HB PDX model and primary patient samples have provided additional evidence supporting an important role of RNR M2 subunit switching in HB growth, drug response, and

relapse. We noticed that the relapse-delaying effect of MK1775 was limited in the HB214 PDX model as all tumors eventually grew back. We used low dose MK1775 to avoid additional drug toxicity and to test its potential use as a safe maintenance therapy. Future studies will be needed to fine tune the dosing schedule of the combinatorial treatment to achieve better prevention of HB relapse. Our additional test on *TP53*-mutant HCC cells suggests that the induction of RRM2B in liver cancer cells is likely p53-dependent, as we did not observe an RRM2 to RRM2B switching in *TP53*-mutant HCC cell lines. However, we did notice a further elevation of RRM2 in HCC cells treated with sorafenib or gemcitabine, suggesting there is a different drug-induced RNR dynamics in *TP53*-mutant tumors. It will be worth further investigation to molecular dissect RNR regulation in *TP53*-

wildtype and mutant liver tumors. There is also more work to be done to explain how these two RNR M2 subunits, when share high homology in their protein sequences, have such distinct functions in HB cells. Our study also suggests that RRM2B-specific inhibitors, if can be developed, will be promising anti-cancer agents to be combined with standard treatment for its critical role in drug resistance and relapse of cancer cells but the non-essential role in normal cells.

HB is one of the most genetically simple cancer types with very low numbers of genomic abnormalities[42]. Little is known how some HB tumors, with minimal alterations in their genomic profiles, manage to progress into highly advanced stages and develop drug resistance. Studies have shown that the drug resistance of many adult solid tumors can be mediated by an adaptive and often reversible cellular state tumor cells turn on under stressful conditions[5,6,43,44]. Past HB studies have mainly focused on the "natural" biology driving advanced tumor progression in an effort on finding better treatment for the high-risk forms of this disease that are resistant to standard treatment[32,45,46]. With the increasing appreciation of cancer cell plasticity in drug resistance through studies in adults, understanding mechanisms that enable HB cells to survive drug treatment and support subsequent relapse has become a potential path leading to a better treatment of this pediatric cancer.

## Methods

**Mice.** Animal protocols were approved by the St. Jude Animal Care and Use Committee. All mice were maintained in the Animal Resource Center at St. Jude Children's Research Hospital (St. Jude). RRM2- and RRM2B- manipulated HepG2 cells were surgically injected into the liver of two-month-old male and female NSG (NOD scid gamma) mice (JAX) at $5 \times 10^4$/mouse in 2 μl cold growth factor-reduced (GFR) matrigel (Corning, Corning, New York) using a 5-μl Hamilton syringe and a 27-gauge needle (Hamilton Company, Bonaduz, Switzerland). Animal survival curves and their median survival were determined by the Kaplan–Meier method in GraphPad Prism 7.

**Cell lines.** HepG2 cells were purchased (HB-8065, American Type Culture Collection, Manassas, Virginia). HB214 cells are a gift from Dr. Stefano Cairo (Xen-Tech, Paris, France).

**HB patient samples.** The de-identified HB patient samples were obtained under a protocol approved by the Institutional Review Boards at St. Jude Children's Research Hospital and Cincinnati Children's Hospital. The informed consent was obtained.

**PDX establishment and in vivo drug efficacy test.** HB214 PDX establishment and drug efficacy studies were performed as described previously[31,47]. HB214 tumor fragments were engrafted in the interscapular region of 6–12-week-old female athymic nude mice (Athymic NudeFoxn1nu, ENVIGO, Gannat, France). After latency period, mice bearing tumor of 62–256 mm³ were randomly assigned to each treatment arm according to their tumor volume to obtain homogenous mean and median tumor volume in each arm. The control group was not treated during all the course of the experiment. Irinotecan HCl trihydrate (MedChem, Monmouth Junction, New Jersey) (dissolved in 99% NaCl 0.9%; 2.5 mg/kg) was administrated intraperitoneally daily for five consecutive days per week for three weeks or first for three consecutive days then one day off followed by five consecutive days per week for two weeks. MK-1775 (MedChem) (dissolved in 0.5% Methylcellulose; 60 mg/kg) was administrated orally three time per week for three or four weeks. Treatments were stopped prematurely if toxicity causing >15% body weight loss was observed. Tumor volumes were measured two to three time per week depending on the tumor growth. Tumors diameters (length and width) are measured with a caliper (digimatic Solaire, IP67) and tumor volume (TV) is calculated using the formula TV (mm³) = [length (mm) × width (mm)²]/2, where the length and the width are the longest and the shortest diameters of the tumor measured perpendicularly, respectively. All animals were weighed at tumor measurement time and were observed every day for physical, behavior, and clinical signs. Control and drug-treated tumors were collected for standard protein and paraffin block preparation.

**RRM2 and RRM2B overexpression.** pLVX-IRES-tdTomato-FlagAkt1 vector was obtained from Addgene (#64831) and flagakt1 was removed creating pLVX-IRES-tdTomato (RTD) control vector. RRM2 and RRM2B fragments were generated using HepG2 endogenous DNA and PCR amplified using Clone Amp HiFi (Takara

#639298). Fragments were purified on a DNA gel and extracted (Qiagen #28706), fragments were then inserted into the RTD control vector creating pLVX-IRES-tdTomato-RRM2 and RRM2B respectively. Plasmids were expressed in E. Coli DH5α competent cells (NEB #C2987H) and purified (Qiagen #12165). Purified plasmid was sent for sanger sequencing and the verified plasmids were sent to the vector core for viral packaging. HepG2 cells were transduced with lentiviral particles at a M.O.I of 3 and the FASC sorted for tdTomato expression.

**RRM2 knockdown.** Lentivirus particles were ordered and added to HepG2 cells at a MOI of 3 (Dharmacon V3SH7669-226296450 (RRM2) VSC6544 (GAPDH), and VSC6572 (Scramble)). Target sequence for RRM2 was (5′ – AGAACCCATTTGA CTTTAT – 3′). Cells were selected using puromycin at a concentration of 2 μg/ml until cells in control wells were dead. A doxycycline dose curve was performed to determine optimal dose of doxycycline without loss of cell viability. 500 ng/ml of doxycycline was the optimal dose, and all future experiments were performed under that concentration.

**RRM2 and RRM2B knockout by CRISPR/Cas9.** $RRM2B^{KO}$ cells were generated using CRISPR/Cas9 technology. Briefly, $4 \times 10^5$ HepG2 cells were transiently transfected with precomplexed ribonuclear proteins (RNPs) consisting of 100 pmol of chemically modified sgRNA 5′ – UUCAUUUACAAUUCCAUCAC- 3′, Synthego, 35 pmol of Cas9 protein (St. Jude Protein Production Core), and 500 ng of pMaxGFP (Lonza) via nucleofection (Lonza, 4D-Nucleofector™ X-unit) using solution P3 and program EN-158 (HB214) or CA-138 (HepG2) in a 20 μl cuvette according to the manufacturer's recommended protocol. Five days post nucleofection, cells were single-cell sorted by FACs to enrich for transfected GFP⁺ cells, clonally selected, and verified for the desired targeted modification (out-of-frame indels) via targeted deep sequencing using gene specific primers with partial Illumina adapter overhangs (F– 5′ TCCATAGTTTACTGGTAGTGGGAT-3′ and R – 5′ AGACATCTTGTCTTTGGCTGAATTT-3′) as previously described[48]. Briefly, approximately $1 \times 10^4$ cells were lysed and used to generate amplicons flanking the gRNA cut site with partial Illumina adapters in PCR #1. During PCR #2 amplicons were indexed and pooled with other amplicons to create sequence diversity. Additionally, 10% PhiX Sequencing Control V3 (Illumina) was added prior to running the sample on an Miseq Sequencer System (Illumina) to generate paired $2 \times 250$ bp reads. Samples were demultiplexed using index sequences and fastq files were generated. NGS analysis of clones was performed using CRIS.py[49]. Knockout clones were identified as clones containing only out-of-frame indels. $RRM2^{KO}$ was attempted in HepG2 $RRM2B^{OE}$ cells using the same protocol (sgRNA 5′ – CG GUCUUGCUGGCCAGGA – 3′). Final clones or pools were authenticated using the PowerPlex® Fusion System (Promega) performed at the Hartwell Center for Biotechnology at St. Jude Children's Research Hospital. Final clones tested negative for mycoplasma by the MycoAlert™ Plus Mycoplasma Detection Kit (Lonza).

**Immunohistochemistry.** Liver and tumor tissues were fixed in neutral buffered formalin for one day at room temperature and submitted to HistoWiz Inc. (Brooklyn, NY) for paraffin processing and embedding. Paraffin sections were cut at 4 μm and analyzed for direct fluorescence microscopy, H&E staining, IHC, and RNAscope[32]. IHC was performed based on the standard protocol. Antibodies used included anti-Ki67 (Abcam, Cambridge, MA, USA, ab16667, 1:200)

**RNAscope staining.** RNAscope in situ hybridization of RRM2 and RRM2B mRNA transcripts was performed on freshly cut paraffin sections according to the manufacturer's protocol (Advanced Cell Diagnostics, Hayward, CA, USA).

**Quantitative RT-PCR.** The total RNA in cells were extracted using RNeasy® Mini Kit (#74106 Qiagen) and the concentration and purity of the RNA were measured by nanodrop spectrometer. SuperScript® III First-Strand Synthesis SuperMix for qRT-PCR (#11752-250 Invitrogen) was used for cDNA synthesis from 1 μg total RNA. FastStart Universal SYBR® Green Master (ROX) (#0491385001 Roche, Mannheim, Germany) was used to perform the quantitative PCR assay on the 7900HT Sequence Detection System (Applied Biosystems, Carlsbad, CA, USA). The results were analyzed using $2^{-\Delta\Delta Ct}$ Method, With ATCB as the internal reference gene. The primers were as follows: RRM2: CTGGAAGGAAAGACTAACTTCTT (Forward), CGTGAAGTCAGCATCCAAGG (Reverse); RRM2B: CCTGCGATG-GATAGCAGATAG (Forward), GCCAGAATATAGCAGCAAAAGATC (Reverse); ATCB: GTTGTCGACGACCAGCG (Forward), GCACAGAGCCTCGCCTT (Reverse).

**Immunoblotting and quantification.** Cells were lysed using radio-immunoprecipitation assay (RIPA) buffer (Thermo Fisher Scientific, Waltham, MA, Cat #89900) supplemented with protease and phosphatase inhibitors (Thermo Fisher Scientific, #78440) and 0.5 M Ethylenediaminetetraacetic acid (EDTA) (#78440, Thermo Fisher Scientific). Lysates were centrifuged at 14,000 rpm for 15 min at 4 °C. Protein concentrations were determined using Pierce™ BCA Protein Assay Kit reagent (Thermo Fisher Scientific, #23227) and separated by electrophoresis on NuPAGE™ 4 to 12% Bis-Tris, 1.0, Protein gel (Invitrogen, Waltham, MA). Antibodies were added according to the manufacturers

recommended conditions. Antibodies used include anti-p53R2 + RRM2 antibody (abcam, Cambridge, United Kingdom, ab209995, 1:10000), anti-RRM2 antibody (abcam, ab172476, 1:5000), phospho-NF-κB p65 (Ser536)(93H1) Rabbit mAb (CST #3033, 1:1000), NF-κB p65(D14E12)XP Rabbit antibody (CST #8242, 1:1000), Phospho-AMPKα (Thr172) antibody (CST #2531, 1:1000), p53 (7F5) Rabbit mAb(CST #2527, 1:1000). Equal protein loading was confirmed using anti-vinculin antibody (abcam, ab219649, 1:5000). Molecular weight marker EZ-Run$^{TM}$ prestained protein ladder (Thermo Fisher Scientific, BP3603500) was used to confirm the expected size of the proteins of interest. Immunoblots were developed with SuperSignal$^{TM}$ West Femto Maximum Sensitivity Substrate (# 34095, Thermo Fisher Scientific) and imaged on a LiCor ODYSSEY Fc (LiCor Inc. Lincoln, NE, Model # 2800). Protein bands were quantified using Image Studio™ Acquisition Software (LI-COR) and normalized to the control bands.

**Cell growth assay**. cells were seeded in a 6-well plate (Day 0) and allowed 24 h to settle and adhere. After 24 h plates were imaged on a Lionheart FX (Biotek) where a protocol was setup enabling us to image the exact same location on days 1 through 5 to monitor growth. Image J was used to measure cell surface area and determine fold change. Cells were seeded in triplicate and the experiment was repeated three times.

**In vitro drug test and cell viability assay**. HepG2 or HB214 (1500 cells/well) was seeded on a 384 well plate in 30 μl. Drugs were added after 24 h. Cell Titer Glo (CTG) viability assay was conducted after 72 h of drug treatment. Briefly, CellTiter-Glo 2.0 (#G9243 Promega) was added to each well in a 1:1 v/v ratio. Plates were then covered to keep away from light and incubated at RT on an orbital shaker at 150 RPM for 30 mins. After the incubation, plate was read on a synergy H4 plate reader for luminescence. Dose effect curves for each drug were calculated using Prism software, version 9 (GraphPad). For drug combinations, responses were analyzed using SynergyFinder2.0[50]. Drugs used include cisplatin (#479036-5 G, Sigma Aldrich), gemcitabine (#AC456890010, Fisher Scientific), vincristine (#AAJ60907MA, Fisher Scientific), triapine (#50-136-4826, Fisher Scientific), MK1775 (#M4102, LKT laboratories, Saint Paul, Minnesota), doxorubicin (#BP25161, Fisher Scientific), sorafenib (#NC0749948, Fisher Scientific), SN-38 (#S4908-50MG, Selleck Chemicals, Harris County, Texas), deferoxamine mesylate (#AC461770010, Acros Organics, Geel, Belgium), KU60019 (#S1570, Selleck Chemicals). All concentrations were seeded in triplicate and the experiment was repeated three times. Significance was determined using the Extra Sum of Squares $f$ test.

**Clonogenic assay**. HepG2 or HB214 ($5 \times 10^5$ /well) cells were seeded in a 6-well plate and incubated for 24 h. The Medium was replaced with medium containing treatment in the concentration stated in the different experiments and the respective controls. After incubation for 72 h cells were trypsonized and seeded ($1 \times 10^4$/well) in a new 6-well plate with fresh medium and allowed to incubate for 14 days. On day 14 wells were washed with PBS and incubated with 6% Glutaraldehyde (#BP2547-1 Fisher) and 0.05% w/v crystal violet (#C581-25 Fisher) for 30 min. After incubation wells were washed with ddH$_2$O and allowed to dry. Plates were then imaged on an Epsom V850 scanner and Image J was used to determine the cell area.

**RNA extraction, sequencing, and data analysis**. The total RNA was extracted from HepG2 cells of different conditions using RNeasy® Mini Kit (#74106 Qiagen) Kit following the manufacturer's protocol. The TruSeq Stranded mRNA LTSample Prep Kit (Illumina) was used for library preparation, and PE-100 sequencing was performed using an Illumina HiSeq X Ten instrument (Illumina). All relevant sequencing data will be available at GEO. The adapters used in library preparation were identified by FastQC (v-0.11.5) (https://www.bioinformatics.babraham.ac.uk/projects/fastqc/) and trimmed from the raw reads by cutadapt (v-1.13) (https://doi.org/10.14806/ej.17.1.200) with the default parameters. RSEM (v-1.3.0, PMID: 21816040), coupled with Bowtie2 (v-2.2.9, PMID: 22388286), were used to quantify the expression of genes based on the reference genome hg38 (GRCh38) with gene annotation from GENCODE (release v32). The Transcripts Per Million (TPM) values were extracted and further transformed to log2(TPM + 0.1) for subsequent analysis. The differential expression analysis was conducted using limma R package (v-3.42.2, PMID: 25605792). The gene set enrichment analysis (GSEA) was performed by the fgsea R package (v-1.12.0) (https://doi.org/10.1101/060012) with MSigDB dataset (v-6.1, PMID: 16199517) and visualized by NetBID software (v-2.0.2). To evaluate the accuracy of gene expression quantification, Salmon (v0.9.1) was employed to calculate the TPM values of genes. The Spearman correlation coefficient and $P$-value were calculated from the TPM values of genes co-identified by RSEM and Salmon using the stats R package (v3.6.1).

**Hub gene identification of RRM2 and RRM2B in HB and HCC**. We used a scalable software for gene regulatory network reverse-engineering from big data, SJARACNe (v-0.1.0, PMID: 30388204), to reconstruct context-dependent signaling interactomes from the gene expression profiles of 46 HB patient samples collected from GSE75271 (PMID: 27775819) and 374 HCC patient samples collected from TCGA-LIHC (PMID: 28622513), respectively. The parameters of the algorithm were configured as follows: $p$ value threshold $p$ = 1e-7, data processing inequality (DPI) tolerance $\epsilon$ = 0, and number of bootstraps (NB) = 200. We used the adaptive

partitioning algorithm for mutual information estimation. Both the upstream and downstream first neighbors of RRM2 or RRM2B were extracted and considered as the hub genes in each context. The whole list of hub gene of RRM2 and RRM2B in HB and HCC were listed in Supplementary Data 1.

**Gene set enrichment analysis of RRM2 and RRM2B hub genes**. To identify the cellular processes regulated by RRM2 and RRM2B, we first applied a hypergeo-metric distribution method for the gene set enrichment analysis using the "funcEnrich.Fisher" function from the R package NetBID (v-2.0.2, PMID: 29849151). Only the HALLMARK and KEGG gene sets from the MSigDB database (v-6.1, PMID: 16199517) were used. The p values of the enrichment analysis of both HB and HCC patient cohorts were further combined with the Stouffer method embedded in the "combinePvalVector" function from NetBID. We then picked the top 10 most significantly enriched gene sets by the hub genes of RRM2 and RRM2B respectively and introduced the Gene Set Enrichment Analysis (GSEA) on both HB and HCC primary patient samples and HepG2 cell lines of different conditions. The differential expression analysis of primary patient samples was performed between RRM2/RRM2B-high and -low which were defined as the top 1/3 and bottom 1/3 in each cohort. The visualization was completed by ggplot2 (v-3.3.4, Wickham H (2016). ggplot2: Elegant Graphics for Data Analysis. Springer-Verlag New York. ISBN 978-3-319-24277-4 (https://ggplot2.tidyverse.org/).

**RRM1 co-immunoprecipitation assay**. RRM1 was immunoprecipitated from whole cell extracts with polyclonal antibodies, and the presence of RRM2 and RRM2B in the precipitated fraction was examined by immunoblotting with the same antibodies mentioned above.

**Nucleotide detection via targeted LC/MS**. Cells were cultured in 6-well plates to ~85% confluence and washed with 2 mL ice cold 1X Phosphate-Buffered Saline (PBS). The cells were then harvested in 300 μL freezing 80% acetonitrile (v/v) into 1.5 mL tubes and lysed by Bullet Blender (Next Advance) at 4 °C followed by centrifugation at 21,000 × $g$ for 5 min at 4 °C. The supernatant was dried by speedvac and reconstituted in 7.5 μL of 66% acetonitrile and 2 μL was separated by a ZIC-HILIC column (150 × 2.1 mm, EMD Millipore) coupled with a Q Exactive HF Orbitrap MS (Thermo Fisher) in negative detection mode. Metabolites were eluted within a 45 min gradient (buffer A: 10 mM ammonium acetate in 90% acetonitrile, pH = 8; buffer B: 10 mM ammonium acetate in 100% H$_2$O, pH = 8). The MS was operated by a full scan method followed by targeted selected ion monitoring and data-dependent MS/MS (tSIM/dd-MS2). MS settings included full scan (120,000 resolution, 350–550 m/z, 3 × 106 AGC and 50 ms maximal ion time), tSIM scan (120,000 resolution, 1 × 105 AGC, 4 m/z isolation window and 50 ms maximal ion time) and data-dependent MS2 scan (30,000 resolution, 2 × 105 AGC, ~50 ms maximal ion time, HCD, Stepped NCE (50, 100, 150), and 10 s dynamic exclusion). Data were quantified using Xcalibur software (Thermo Fisher Scientific) and normalized by cell numbers. Ribonucleotide and deoxyribonucleotides were validated by authentic standards.

**Statistics and reproducibility**. All measurements were taken from ≥ three distinct samples. Experimental data were analyzed using the unpaired two-tailed Students $t$ test. Drug response curves were analyzed with GraphPad software using the Extra Sum of Squares $f$ test. Kaplan Meier curves for survival were analyzed with GraphPad using the log-rank test.

**Reporting summary**. Further information on research design is available in the Nature Portfolio Reporting Summary linked to this article.

## Data availability
The RNAseq datasets generated during this study are available in the NCBI GEO repository (GSE223839). The numerical numbers of the graphs are included in Supplementary Data 2. Blot/gel image quantifications are included in Supplementary Data 3. Uncropped and unedited blot/gel images are included as Supplementary Fig. 17. The LC-MS nucleotide metabolite data and the *RRM2* and *RRM2B* cDNA plasmids are freely available upon request. The authors confirm that the data supporting the findings of this study are available within the article and its supplementary materials, or available from the corresponding author on reasonable request.

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

## Acknowledgements
We thank NIH/NCI for the funding support (R21CA256464).

## Author contributions
A.B. conducted most of the biological experiments. Q.P. conducted the computational analysis. E.I. conducted in vivo drug testing. C.T., L.F., L.L., B.S.H., H.T., and M.F. conducted experiments. N.T., S.M.P., J.P., J.Y., and S.C. provided technical and intellectual support. L.Z. conceived and oversaw the research. All authors contributed to the writing of the manuscript.

## Competing interests
The authors declare no competing interests.
