## [Peer Review File · Communications Biology]

Reviewers' comments:

Reviewer #1 (Remarks to the Author):

Brown et al., Comms Biology 2022

Here the authors as describe in hepatoblastoma that the RRM2 subunit of RNR is associated with aggressive disease and proliferation but that in response to stress (chemotherapy) subunit switching occurs to favour the less active RRM2B subunit. During recovery after stress the RNR composition reverts back to RRM1+ RRM2. The data presented includes patient data which largely supports the conclusions drawn. However, the findings lack novelty. The key points made are reported in the literature including RRM2 being the more active subunit, RRM2B being a p53 target, RRM2B being the stress responsive small subunit, increased RRM2 occurring in tumours. Despite this my major concern is the use of MK1775 as a RRM2 inhibitor with no mention of its impact on Wee1 and the cell cycle.

Specific points

The data in figure 3G is referred to as not surprising but it is unclear why this is the case. Previous figures have argued that RRM2 is associated with aggressive disease and increased proliferation. The authors seem to present these first data in a positive/significant way but then argue they are not biologically significant in response to figure 3G. This is somewhat confusing to the reader and more measured description of the data would help.

In parts H/I the authors indicate that RRM2oe cells have more Ki67+ve cells but do not relate this back to their finding that there is no difference in growth rate between the tumour types – how do they explain increased Ki67 with no increase in proliferation rate? Surely this means the Ki67 finding is not biologically significant or suggests another role for Ki67 in this model?

MK1775 is a Wee1 inhibitor.

Evidence is required to demonstrate that triapine and MK1775 is inhibiting RRM2 (change in nucleotide level)

Data to show the impact of both 'RRM2' inhibitors on the cell cycle also needed. As pointed out by the authors, RRM2 is cell cycle regulated and blocking Wee1 will impact the cells cycle.

Why is the increase in RRM2B interesting (line 192)? RRM2B is stress responsive, the authors have induced stress (line 182).

Reference 32 (line 202) refers to p53R2 which is RRM2B – as far as I know there is no p53-dependency to RRM2 expression, if it was p53-regulated how would this fit with the cell cycle dependent regulation observed?

Minor points

Some English language editing required, can authors confirm they mean multidisciplinary therapies in line 75?

An alternative way of presenting figure 2 should be sought as the gene names are not legible at publication size.

Reviewer #2 (Remarks to the Author):

The manuscript by Brown et al. entitled 'Ribonucleotide reductase subunit switching in hepatoblastoma drug response and relapse' describes an oncogenic role of RRM2 in HB. Furthermore, the authors provide interesting insights into a RRM2-RRM2B switch to establish post-therapy survival and relapse, after which the cells gradually switch back to RRM2 dependency. In this framework, they provide evidence for a potential therapeutic vulnerability of simultaneous targeting RRM2 in combination with chemotherapy. These findings are highly interesting and will be interesting to a broad audience. However, there are several minor concerns that should be addressed before publication.

1. In figure 3, it would be of added value to further characterize the molecular responses eg for RRM2 shRNA transduced cells as well as to functionally assess better the impact on replication stress (using eg DNA combing or western blot analyses)
2. In the paper, the authors state the use of MK1775 as an RRM2 inhibitor, but this actually is indirect through pharmacological WEE1 inhibition. I would propose to rephrase this in the manuscript. Additionally, the in vivo experiments in figure 7 were done with irinotecan and MK1775 due to toxicities observed with triapine. Can these experiments be repeated with the inclusion of triapine by lowering those doses?
3. Why the focus on combination therapy with chemotherapeutics and not more on other available targeted drugs? Would the described RRM2-RRM2B switch be specific to chemotherapy exposure or would this be a general phenomenon? Can this be further shown with more on-target drugs?
4. The proposed RRM2-RRM2B switch is a very intriguing finding, but during relapse/drug resistance most of the tumors indeed get p53 independent and then it would be highly valuable to what the dependency is switching to. Can this be approached through a generalized proteomics screening analysis or other?
5. It is highly recommended to provide proper quantification of all the Western blot data included.

Liqin Zhu, PhD

Assistant Member

Division of Pharmaceutical Sciences
Department of Pharmacy and Pharmaceutical Sciences
St. Jude Children's Research Hospital
262 Danny Thomas Place
I5304A, MS 313, Chili's Care Center
Memphis, TN 38105-3678
Phone: (901) 595-5250

January 19th, 2023

Dear *Oncogene* Editors,

We appreciate *Communications Biology*'s review of our manuscript entitled, "Ribonucleotide Reductase Subunit Switching in Hepatoblastoma Drug Response and Relapse" (COMMSBIO-22-2653-T), and your invitation to submit a revised manuscript. We have carefully considered all of the reviewer comments, performed additional analyses to address each of the questions/critiques as outlined below (marked in red), and have modified the manuscript accordingly (marked in red in the text and highlighted in yellow in the figures). For ease of evaluation, we have described our responses directly within the template of the reviewers' comments.

Reviewer 1:

Here the authors as describe in hepatoblastoma that the RRM2 subunit of RNR is associated with aggressive disease and proliferation but that in response to stress (chemotherapy) subunit switching occurs to favour the less active RRM2B subunit. During recovery after stress the RNR composition reverts back to RRM1+ RRM2. The data presented includes patient data which largely supports the conclusions drawn. However, the findings lack novelty. The key points made are reported in the literature including RRM2 being the more active subunit, RRM2B being a p53 target, RRM2B being the stress responsive small subunit, increased RRM2 occurring in tumours. Despite this my major concern is the use of MK1775 as a RRM2 inhibitor with no mention of its impact on Wee1 and the cell cycle.

Specific points:

1. The data in figure 3G is referred to as not surprising but it is unclear why this is the case. Previous figures have argued that RRM2 is associated with aggressive disease and increased proliferation. The authors seem to present these first data in a positive/significant way but then argue they are not biologically significant in response to figure 3G. This is somewhat confusing to the reader and more measured description of the data would help. In parts H/I the authors indicate that RRM2oe cells have more Ki67+ve cells but do not relate this back to their finding that there is no difference in growth rate between the tumour types – how do they explain increased Ki67 with no increase in proliferation rate? Surely this means the Ki67 finding is not biologically significant or suggests another role for Ki67 in this model?

We thank the reviewer for this question and we apologize for the seemingly conflicting statements on the significance of RRM2 in HB. We did find the level of RRM2 associates with HB tumor progression in both our animal model and in patients via multiple approaches as shown in **Figure 1**. However, the HB cell line we used in **Figure 3**, HepG2, is an aggressive cell line with a fast growth rate both in vitro and in vivo. In an unrelated, unpublished study in our lab, we performed an RNAseq

analysis of HepG2 and a number of cell lines derived from the HB PDX models reported in Nicolle *et al.* (*Hepatology* 2016;64:1121-35).¹ The result is in the **Response Letter Figure 1** below. It shows the high level of endogenous *RRM2* in HepG2 cells compared to other HB cells derived from HB PDX models which are high-grade tumors in general. We believe that this high level of endogenous *RRM2* is the reason why further overexpression of *RRM2* only had a statistically significant but only modest impact on HepG2 cell proliferation in vitro as shown in **Figure 3B**. After transplanting these cells into mouse liver, we noticed that the cell proliferation was highly heterogeneous in in vivo HepG2 tumors, likely being affected by many environmental factors. Therefore, although “the most proliferative regions of the *RRM2*^{OE} tumors had a significantly higher percentage of Ki67⁺ cells than those of the tdT tumors (**Figure 3H, I**)” (Line 177-178), the modest impact of *RRM2* overexpression on HepG2 cells did not translate to a significant difference in the survival of mice transplanted with tdT control vs. *RRM2*^{OE} HepG2 cells as shown in **Figure 3G**.

We apologize that we did not clearly explain the difference between the impact of *RRM2* overexpression in cell proliferation vs. animal survival in the original manuscript. Therefore, we added this following statement in the **Revised manuscript, Line 175-177**: “When examining tumor proliferation via Ki67 IHC, we noticed that the cell proliferation was highly heterogeneous in HepG2 tumors in vivo, likely being affected by many environmental factors.”

2. MK1775 is a Wee1 inhibitor. Evidence is required to demonstrate that triapine and MK1775 is inhibiting *RRM2* (change in nucleotide level). Data to show the impact of both ‘*RRM2*’ inhibitors on the cell cycle also needed. As pointed out by the authors, *RRM2* is cell cycle regulated and blocking Wee1 will impact the cells cycle.

We sincerely apologize that we did not clarify that MK1775 is designed as a WEE1 inhibitor. We used MK1775 in our study as an *RRM2* inhibitor in addition to triapine for three reasons: (1) triapine, the most widely used *RRM2* inhibitor, induces rather than inhibits *RRM2* in HepG2 and HB214 cells at lower concentrations, and only shows *RRM2*-inhibiting effect at concentrations $\geq 6.25 \mu\text{M}$ (**Figure 4B**); (2) in vivo testing of triapine in the HB214 PDX model shows high drug toxicity; (3) Pfister *et al.* (*Cancer Cell* 2015;28:557-568)² shows that WEE1 inhibition by MK1775 selectively kills H3K36me3-deficient cancer cells through dNTP starvation resulting from *RRM2* depletion. This article was cited in our original manuscript and our in vitro data also demonstrate an efficient inhibition of *RRM2* by MK1775 (**Figure 4B**). Nevertheless, MK1775 is not an *RRM2* inhibitor and we have made the following modification when introducing the two drugs in the **Revised manuscript**: “We tested an *RRM2* inhibitor, triapine²⁷ and a WEE1 inhibitor MK1775 which has been shown to inhibit cell cycle via depleting *RRM2*²⁸...” (Line 184-185).

To address the reviewer's comment on the effect of RRM2 inhibition on nucleotide level as well as cell cycle, we performed the following experiments included in the **Response Letter Figure 2** on Page 4:

- (1) Nucleotide detection via targeted liquid chromatography/mass spectrometry in HepG2 cells treated with triapine and MK1775. We detected consistent reduction in the nucleotide levels in the cells treated with triapine and MK1775 compared to the control cells (**Response Letter Figure 2A**);
- (2) Immunoblotting of the cell cycle marker p21 and DNA damage marker γ -H2AX in HepG2 cells treated with triapine and MK1775 at the various concentrations. We noticed marked induction of p21 and γ -H2AX associated with the reduction of RRM2 protein level (**Response Letter Figure 2B**).

These results are also added to the **Revised Manuscript** as **Supplemental Figure S6, Line 190-193**: **“Consistent with their RRM2-inhibiting function, triapine and MK1775 treatment led to a significant reduction in the nucleotide levels in HepG2 cells along with a strong induction of the cell cycle suppressor p21 and the DNA damage marker γ -H2AX (Supl Figure S6).”**

3. Why is the increase in RRM2B interesting (line 192)? RRM2B is stress responsive, the authors have induced stress (line 182).

We thank the reviewer for this comment. RRM2B, indeed, has been demonstrated as a stress response. However, its induction has mostly been demonstrated in endogenous stress conditions in tumors, such as hypoxia, oxidative stress, and senescence. RRM2B upregulation has not been reported in related to RRM2 inhibition or to chemotherapy to our knowledge. Therefore, RRM2B induction we observed in our HB cells and tumors in response to chemotherapy, while in line with the previous knowledge that it involves in stress response, is a new discovery that contributes to HB chemoresistance as our study demonstrated. To explain why we thought it was an interesting observation, we added this statement in the **Revised Manuscript (Line 196-197)**: **“RRM2B upregulation has not been reported in the context of RRM2 inhibition or to chemotherapy.”**

4. Reference 32 (line 202) refers to p53R2 which is RRM2B – as far as I know there is no p53-dependency to RRM2 expression, if it was p53-regulated how would this fit with the cell cycle dependent regulation observed?

We thank the reviewer for pointing out this mistake. “RRM2” in Line 202 in the **Original Manuscript** should be “RRM2B”. It has been corrected in the **Revised Manuscript**.

Minor points:

1. Some English language editing required, can authors confirm they mean multidisciplinary therapies in line 75?

We thank the reviewer for this comment. We have included this statement in the **Revised Manuscript (Line 76)**: ...with multidisciplinary therapies **“including surgery, chemotherapy, and radiotherapy”**. We have also had our scientific editing team check our manuscript for language.

2. An alternative way of presenting figure 2 should be sought as the gene names are not legible at publication size.

We agree with the reviewer on this comment. We have modified **Figure 2A & 2C** in the **Revised manuscript** to increase the font size of the RRM2 and RRM2B hub genes. The revised figure is also included as the **Response Letter Figure 3** on page 5.

Response Letter Figure 2. RRM2 inhibition in HepG2 cells leads to nucleotide reduction, cell cycle arrest and DNA damage response.

(A) Quantitative analysis of nucleotide levels in HepG2 cells treated with control (ctrl, DMSO), triapine (Tri, 3.125 mM), and MK1775 (MK, 0.39 mM) via targeted liquid chromatography/mass spectrometry (biological replicates: n=5 per group).

(B) Immunoblotting of the indicated proteins in HepG2 cells treated with triapine and MK1775 at the indicated concentration.

Response Letter Figure 3. A modified version of Figure 2 in the Revised manuscript.

Reviewer 2:

The manuscript by Brown et al. entitled ‘Ribonucleotide reductase subunit switching in hepatoblastoma drug response and relapse’ describes an oncogenic role of RRM2 in HB. Furthermore, the authors provide interesting insights into a RRM2-RRM2B switch to establish post-therapy survival and relapse, after which the cells gradually switch back to RRM2 dependency. In this framework, they provide evidence for a potential therapeutic vulnerability of simultaneous targeting RRM2 in combination with chemotherapy. These findings are highly interesting and will be interesting to a broad audience. However, there are several minor concerns that should be addressed before publication.

1. In figure 3, it would be of added value to further characterize the molecular responses eg for RRM2 shRNA transduced cells as well as to functionally assess better the impact on replication stress (using eg DNA combing or western blot analyses).

We thank the reviewer for this comment. This is a very similar suggestion as Reviewer 1’s Comment #2. Unfortunately, our doxycycline-inducible RRM2 shRNA virus has been frozen for over five years and we found it had lost its activity. However, as shown in the **Response Letter Figure 2** on Page 4 in our response to Reviewer 1’s Comment #2, we performed the following experiments to show that RRM2 inhibition by triapine and MK1775 led to consistent reduction in nucleotide levels, induction in cell cycle suppressor p21 and DNA damage marker γ -H2AX:

- (1) Nucleotide detection via targeted liquid chromatography/mass spectrometry in HepG2 cells treated with triapine and MK1775. We detected consistent reduction in the nucleotide levels in the cells treated with triapine and MK1775 compared to the control cells (**Response Letter Figure 2A**);
- (2) Immunoblotting of the cell cycle marker p21 and DNA damage marker γ -H2AX in HepG2 cells treated with triapine and MK1775 at the various concentrations. We noticed marked induction of p21 and γ -H2AX associated with the reduction of RRM2 protein level (**Response Letter Figure 2B**).

These results are also included in the **Revised Manuscript** as **Supplemental Figure S6**.

2. In the paper, the authors state the use of MK1775 as an RRM2 inhibitor, but this actually is indirect through pharmacological WEE1 inhibition. I would propose to rephrase this in the manuscript.

We thank the reviewer for pointing it out. This is also a similar concern to Reviewer 1’s Comment #2. Please refer to our response on Page 2 of the response letter, and we have made the following modification when introducing the two drugs in the **Revised manuscript**: “**We tested an RRM2 inhibitor, triapine²⁷ and a WEE1 inhibitor MK1775 which has been shown to inhibit cell cycle via depleting RRM2²⁸...**” (Line 183-184).

Additionally, the in vivo experiments in figure 7 were done with irinotecan and MK1775 due to toxicities observed with triapine. Can these experiments be repeated with the inclusion of triapine by lowering those doses?

We thank the reviewer for this suggestion. We have included the in vivo data on triapine treatment in the **Response Letter Figure 4** on the next page. As it shows, triapine did not have a strong tumor-suppressing effect at the concentration we tested that already caused significant toxicity. Therefore, we hope the reviewer will agree that triapine at a lower dose would most likely have an even less effect on HB214 tumor growth in vivo, therefore, is difficult to demonstrate the potential of inhibiting RRM2 as a maintenance treatment to prevent HB relapse. We suspect that the formulation of triapine we purchased was the cause of toxicity and are working with our drug treatment team to improve its in vivo performance.

3. Why the focus on combination therapy with chemotherapeutics and not more on other available targeted drugs? Would the described RRM2-RRM2B switch be specific to chemotherapy exposure or would this be a general phenomenon? Can this be further shown with more on-target drugs?

We thank the reviewer for this comment. We focused on chemotherapy because HB tumors are very chemosensitive in general, therefore, chemotherapy is still the first-line treatment for HB. Targeted therapies have not been widely used in HB patients. With this being said, we agree with the reviewer that it is interesting to find out if RRM2-RRM2B switch is specific to chemotherapy or a general response of HB cells to drug treatment. We have included the following two drugs in the **Response Letter Figure 5** on the next page to address this question:

- (1) In the **Supplemental Figure S14** in the **Original Manuscript**, we have shown that an ATM inhibitor, KU60019, does not induce changes in RRM2 or RRM2B protein level in HepG2 cells while capable of killing HepG2 at the concentrations tested. These results are copied to the **Response Letter Figure 5**.
- (2) We performed a similar drug test using a multi-targeted tyrosine-kinase inhibitor, ponatinib. HepG2 showed high sensitivity to ponatinib (**Response Letter Figure 5A**). However, immunoblotting of RRM2 and RRM2B did not show a switch from RRM2 to RRM2B associated with cell death (**Response Letter Figure 5B**). It is rather interesting that RRM2 levels dropped at the lower concentrations, then increased with increased ponatinib concentration. RRM2B showed no changes in its protein level until at the highest ponatinib concentration used, when RRM2 continued to increase rather than decrease.

Overall, these results suggest that the RRM2-RRM2B switch in HB cells is likely specific to chemotherapy from the evidence we gathered so far. Their responses to more targeted drugs need to be further investigated.

4. The proposed RRM2-RRM2B switch is a very intriguing finding, but during relapse/drug resistance most of the tumors indeed get p53 independent and then it would be highly valuable to what the dependency is switching to. Can this be approached through a generalized proteomics screening analysis or other?

We thank the reviewer for this suggestion. It is indeed a very interesting question that how the relapsed tumor cells may differ to treatment-naïve cells, as it is well known that their drug response is often different from the original tumor. If we could direct the reviewer to our main **Figure 7A**, the reviewer may notice that the relapsed HepG2 cells after cisplatin treatment at 6.25 and 12.5 µM showed lower RRM2 and higher RRM2B than the non-treated cells, or the cells treated with 3.125 µM cisplatin. We found this very interesting as it suggests that HB cells recovered from intense drug treatment may have acquired different mechanisms to support their growth. We have pending grants on using coupled proteomic and metabolomic profiling to dissect the molecular changes in HB cells post high doses of chemotherapies.

5. It is highly recommended to provide proper quantification of all the Western blot data included.

We thank the reviewer for this suggestion. We performed the quantification of all the Western blot data in the main figures and included the results in the source data file: **Brown et al_Source data for figures**.

References:

1. Nicolle D, Fabre M, Simon-Coma M, et al. Patient-derived xenografts from pediatric liver cancer predict tumor recurrence and advise clinical management. *Hepatology* 2016.
2. Pfister SX, Markkanen E, Jiang Y, et al. Inhibiting WEE1 Selectively Kills Histone H3K36me3-Deficient Cancers by dNTP Starvation. *Cancer Cell* 2015;28:557-568.

REVIEWERS' COMMENTS:

Reviewer #1 (Remarks to the Author):

The authors have now included the critical data, which was acknowledgment of the activity of MK1775 and dNTPS measurements. The novelty of RRM2B expression in response to chemotherapy is still questionable (it is a p53 target and p53 is stabilised by chemo) but the text has been modified appropriately.

I suggest publication of revised manuscript.

Reviewer #2 (Remarks to the Author):

I agree with the revised version of the manuscript and the answers provided to my questions.

Liqin Zhu, PhD

Assistant Member
Division of Pharmaceutical Sciences
Department of Pharmacy and Pharmaceutical Sciences
St. Jude Children's Research Hospital
262 Danny Thomas Place
I5304A, MS 313, Chili's Care Center
Memphis, TN 38105-3678
Phone: (901) 595-5250

February 14, 2023

Dear Editors,

We thank *Communications Biology* for accepting our manuscript in principle. Neither of the two reviewers have suggestions for further revisions.

Reviewer 1:

The authors have now included the critical data, which was acknowledgment of the activity of MK1775 and dNTPS measurements. The novelty of RRM2B expression in response to chemotherapy is still questionable (it is a p53 target and p53 is stabilised by chemo) but the text has been modified appropriately.

I suggest publication of revised manuscript.

Reviewer 2:

I agree with the revised version of the manuscript and the answers provided to my questions.

Sincerely,

Liqin Zhu